# Effects of unconditional cash transfers on family processes and wellbeing among mothers with low incomes

Katherine A. Magnuson [1] ✉, Greg J. Duncan [2], Hirokazu Yoshikawa[3], Paul Y. Yoo[4], Sangdo Han[1], Lisa A. Gennetian [5], Sarah Halpern-Meekin [6], Nathan A. Fox[7] & Kimberly G. Noble[8]

This study examines causal impacts of unconditional cash transfers on economic hardship and key family processes that may affect children's development. The study randomized 1000 mothers of newborns, with prior-year household income below the federal poverty threshold, to receive unconditional cash transfers of $333 or $20 per month (Clinical Trial Registry number NCT03593356). Data collected approximately 12, 24 and 36 months after the child's birth show a moderate increase in household income and reductions in poverty; no statistically significant improvements in subjective economic hardship reports or quality of play with infants; and small, mostly statistically non-significant, increases in parental psychological distress and declines in mothers' relationship quality. However, mothers receiving the higher amount reported more frequently engaging in enriching child activities than mothers receiving the lower amount. Cash support may provide other benefits for families and children, but moderate support levels do not appear to address self-reported economic hardship or standard survey measures of maternal well-being. However, these results do not rule out the possibility of very small effects.

Research has shown that poverty experienced during early childhood is associated with worse child, youth, and adult outcomes. Aspects of development affected by poverty early in life include learning, educational attainment, physical and mental health, and adult earnings[1]. The negative effects of poverty may be particularly strong when it is experienced during early childhood, relative to later in childhood and in adolescence[2]. Developmental theory suggests that family processes play an important role in explaining associations between income poverty and children's outcomes[3]. However, the extent to which income support and anti-poverty initiatives causally affect family processes among families with low incomes is not well understood.

Many longitudinal, non-experimental studies have established the cascading effects of economic hardship and low income, which contribute to parental psychological distress and reduce the quality of family relationships, parenting, and child outcomes. The goal of this randomized controlled study is to examine the extent to which a multi-year cash transfer program—which provides an ~$4000 annual unconditional cash transfer, distributed monthly as a cash gift on a debit card to participants who had given birth (henceforth, mothers) and had prior-year household income below the federal poverty threshold—affects economic hardship, maternal well-being, family relationships, and parenting across the first three years of life.

[1]Sandra Rosenbaum School of Social Work, University of Wisconsin, Madsion, WI, USA. [2]School of Education, University of California, Irvine, CA, USA. [3]Steinhardt School of Culture, Education, and Human Development, New York University, New York, NY, USA. [4]Graduate School of Education, Stanford University, Stanford, CA, USA. [5]Sanford School of Public Policy, Duke University, Durham, NC, USA. [6]Department of Human Development and Family Studies, University of Wisconsin, Madison, WI, USA. [7]College of Education, University of Maryland, College Park, MD, USA. [8]Teachers College, Columbia University, New York, NY, USA. ✉e-mail: kmagnuson@wisc.edu

It is well understood that poverty and poor mental health are highly correlated. Ridley et al.[4] summarize the existing evidence from experimental studies of income changes in low- and middle-income countries, concluding that, on average, cash transfers have small positive impacts on mental health, whereas negative economic shocks undermine mental health. The mechanisms by which poverty affects mental health are diverse, including financial uncertainty and worry, exposure to harmful environmental contexts, and experiences of violence, trauma, and crime[4].

Scholars of family systems have further argued that economic hardship resulting from poverty is especially harmful to parents, because of their role in caring for children. The Family Stress Model posits that poverty results in economic hardship, and this in turn generates parental psychological distress and decreases emotional well-being, which has an adverse effect on co-parental relationships and parenting quality. In turn, harsher as well as less warm and stimulating parenting results in worse child and adolescent behavioral and academic outcomes[5–7] (Fig. 1). This model was first developed in the context of the Great Depression and Iowa farm floods and has since been applied to urban families of differing racial and ethnic backgrounds across both early childhood and adolescence[3,7–10].

Numerous correlational studies have estimated path models of the associations from low income and economic hardship to children's developmental outcomes, as depicted in Fig. 1 (see Masarik and Conger[3] for a review). However, rarely has the model been estimated in the context of an anti-poverty program, to examine whether cash supports that reduce family poverty causally improve family processes. A key theoretical question about the Family Stress Model is raised in the context of cash transfer programs. As reviewed by Baranov et al.[11], when cash transfers are provided to a mother, some theories suggest it may increase threats or experiences of intimate partner violence. Status inconsistency and gendered resource theories suggest cash transfers might increase conflict and violence because a woman's increased income threatens the male partner's gendered ideas about resources and status[12]. Household bargaining theory argues that threats and use of violence are an instrumental tool that male partners might use to garner a larger portion of the cash transfer[13]. These theories run counter to the predictions from the Family Stress Model that suggest increased resources should reduce parents' stress and thus improve the quality of their relationships.

Most conditional and unconditional cash transfer programs have been conducted in low- and middle-income countries. Experimental evaluations of these programs typically include some measures of economic well-being as well as physical and mental health, but few include measures of family processes. Systematic reviews of these international studies find that, on average, cash transfer programs improve food security and indicators of economic well-being as well as mental health[14–17]. A recent meta-analysis of 45 conditional and unconditional cash transfer evaluations conducted in low- and middle-income countries found average effect sizes of +0.13 on adult subjective well-being and −0.07 on mental health problems (primarily depressive symptoms)[18]. In addition, most cash transfer programs either reduce intimate partner violence or do not affect it at all. Baranov et al.'s[11] meta-analysis found that cash transfer programs reduce intimate partner violence by 2–4% depending on the type of violence measured.

However, systematic reviews suggest considerable heterogeneity in the estimated impacts of cash transfer programs on parent and child well-being, both across and within studies[15,18]. McGuire et al.'s[18] meta-analysis found that unconditional cash transfers had larger impacts than conditional ones, and that impacts declined over time after the payments ended. Yet not all results are consistent—an evaluation of a unconditional cash transfer program in Ecuador found that monthly cash transfers amounting to 10% of household income improved young children's development among families with the lowest incomes, but generated null to negative effects on maternal mental health among those same families[19,20]. The same cash transfer in Ecuador reduced emotional violence in households for mothers with higher educational attainment, but not for mothers with lower educational attainment[21]. Among mothers with lower educational attainment, if the mother's educational attainment was higher than her partner's, the cash transfer appeared to increase emotional violence[21].

In the United States, studies of the Earned Income Tax Credit (EITC), as well as Child Tax Credit (CTC) expansions and welfare reform, provide more inconsistency in the evidence that poverty reduction policies and programs improve parental well-being and parenting[22–24]. Experimental and quasi-experimental studies of expansions of the EITC and similar programs have found that these benefits have positive impacts on the mental health of women[22,25,26] and the quality of children's home environments[27]. In contrast, a synthesis of welfare reform studies showed that programs that increased mothers' incomes by a few thousand dollars generated only selective reductions in economic hardship and improvements in maternal mental health. In addition, these studies reported null to small impacts on the quality of family relationships and parenting[24]. Yet, it is hard to extrapolate from these studies to unconditional cash transfers because these tax and welfare policies and programs condition increased incomes on employment.

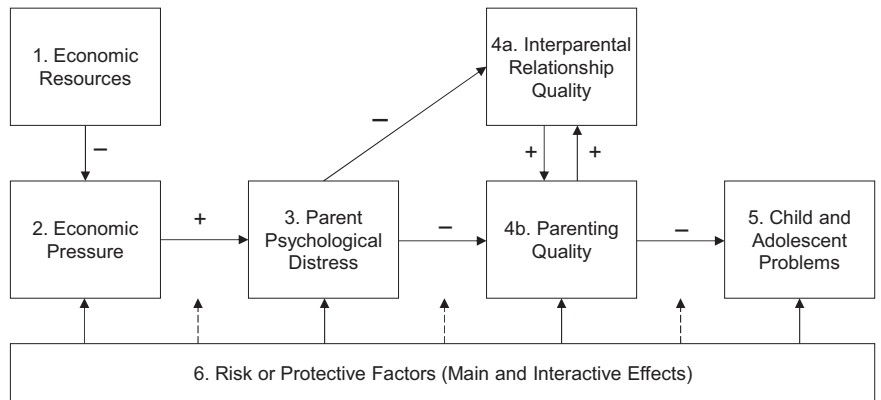

Fig. 1 | Adaptation of the family stress model. Authors' adaptation of the Family Stress Model[3]. The adapted model projects that an increase in (1) economic resources decreases (2) economic pressure in the household. A decrease in economic pressure decreases (3) parent psychological distress, which subsequently increases both (4a) interparental relationship quality and (4b) parenting quality, which are interlinked. Improvements in parenting quality decreases (5) child and adolescent problems. The surrounding environment's (6) risk and protective factors directly affect each component of the model (1–5) and also moderate the relationships between the components (1–5).

Studies of recurring cash transfers to families with low income are rare in the United States and yield mixed results[28]. An evaluation of a conditional cash transfer study conducted in New York City and Memphis found reduced economic hardship and small positive improvements in parents' psychological well-being[29]. More recently, the expansion of the CTC in 2021 provided an opportunity to study the impact of generous cash payments provided to parents for six months. Quasi-experimental studies clearly show that the CTC reduced poverty and multiple aspects of economic hardship[30]. Yet studies also showed mixed findings for parents' mental health outcomes, including depression and anxiety[30–34]. In the case of the CTC, it is also hard to know whether findings would generalize to contexts outside of the COVID-19 pandemic.

Finally, related evidence comes from experimental studies of one-time unconditional cash transfers provided during the COVID-19 pandemic in the United States. These studies found that while the transfers increased spending, they did not have positive impacts on other outcomes, including psychological well-being. Pilkauskas et al.[35] and Jacob et al.[36] studied two rounds of $1000 one-time cash transfers given by GiveDirectly to families receiving food assistance in 12 states. The first study[35] examined the impact of payments that were disbursed in May 2020, and found that while spending increased, the transfers did not change any of the key outcomes considered: material hardship, mental health challenges, partner conflict, child behavior problems, or parenting behaviors. A subsequent study[36] of similar payments provided in September 2020 also reported null effects on self-reports of stress, anxiety, and depression. An additional experimental study[37] provided unconditional cash to individuals who had approached a non-profit organization seeking pandemic cash relief. Individuals were randomly assigned to a control group or to receive a one-time payment of either $500 or $2000. Surprisingly, results indicated that both payments had negative impacts on subjective measures of well-being after several months, which the authors attributed to the windfall of money not being large enough to take care of recipients' needs. Again, a key limitation of these pandemic studies is the uncertainty about whether findings would generalize outside of the unique context created by the global public health crisis, and whether larger or more regular payments would yield differing results.

Nearly all studies of cash transfers have focused on families or adults without regard to the age of their children. As a result, when thinking about how cash transfers might affect families with very young children, several additional factors warrant consideration. First, romantic partner relationships may change in both unmarried and married households following the birth of a child[38]. Early versions of research on the Family Stress Model focused on families with two heterosexual parents, while later work included single-mother families[39]. Owing to considerable complexity and fluidity in family structures, especially in the early years of a child's life, the concept of parental relationship quality may need to be expanded to include the quality of co-parenting relationships with a former partner as well as romantic relationships with partners who are not biological parents of the resident children. Likewise, for families with young children, it may be especially useful to also assess impacts on parenting stress[40,41]. Given the care and attention required in the early years of life, young children make uniquely intensive demands on parents. Even when parents have other children, the birth of a child often necessitates reorganization of responsibilities and roles within families, and this can contribute to stress associated with the parenting role[42].

The Baby's First Years (BFY) study provides a U.S.-based test of the effects of a regular monthly unconditional cash transfer program on economic well-being, the quality of family relationships, maternal mental health and well-being, and parenting stress measured among mothers of 12-, 24-, and 36-month-olds. In this ongoing randomized controlled trial, 1000 mothers with reported income below the federal poverty threshold at the time of recruitment across four U.S.

metropolitan areas were randomly assigned to receive either the "low-cash gift" of $20 per month or the "high-cash gift" of $333 per month. Gennetian et al.[43] found that the unconditional cash transfers in the BFY study had selective impacts on the amount of time and money that mothers invest in their young children. Estimates suggest that mothers spent more time engaged in cognitively stimulating activities with their children. In addition, ~25% of the value of the cash gift was used on children's books, toys, activities, clothing, diapers, and children's electronic items/devices. However, there were no statistically significant impacts of the cash gift on core household expenditures such as food and rent, mothers' participation in paid work or sources of other household income, or children's time in child care. Gennetian et al.'s[43] analysis shows a great deal of diversity in locations of debit cards transactions, including ATMs, big box stores, gas stations, restaurants, children's stores, and phone bills.

In this work, we further explore the impact of the BFY monthly unconditional cash transfers by considering their impact on other indicators of family well-being. Our research questions are: (1) Does a monthly unconditional cash transfer for mothers with low income (prior-year household income below the federal poverty threshold) of infants and toddlers in the United States reduce economic hardship, maternal stress (self-reported and physiological), parenting stress, and mothers' depressive and anxiety symptoms? (2) Does a monthly unconditional cash transfer improve co-parenting and relationship quality with current or past romantic partners? And, finally, (3) does a monthly unconditional cash transfer improve aspects of parenting, including warmth/encouragement, harshness, and stimulating activities?

## Results

Characteristics of mothers at the time of recruitment are found in Table 1. Mothers averaged about 27 years of age and had completed slightly fewer than 12 years of schooling on average. Mothers reported an average annual household income of about $22,000. About 40% of the sample self-identified as non-Hispanic Black, and another 40% self-identified as Hispanic of any race. Close to half of the mothers had never been married, and a little over one-third of the mothers reported living with the biological father of their newborn. About one-third of the newborns were first births (not shown in Table 1 because it was not pre-registered as a baseline covariate). About 24% of mothers lived in households without any other adults and had at least one other child in the household in addition to the focal child (not shown in Table 1 because it was not pre-registered as a baseline covariate). Comparable statistics for the samples at later ages are found in Supplementary Tables 1 to 3.

### Intent-to-treat analyses

Our research questions ask whether receiving an unconditional monthly cash gift of $333 improved measures of key constructs of maternal subjective well-being and parenting (Supplementary Tables 4 and 5) compared with receiving a monthly cash gift of only $20. Tables 2 and 3 provide descriptive information on mean differences in outcomes between the high-cash and low-cash gift groups.

With its $3756 annual cash gift payment difference between the high-cash and low-cash gift groups, BFY was designed to increase economic resources. Although efforts were made to ensure that, to the extent possible, the cash gifts did not count as income in determining eligibility for benefits from most safety net programs, changes in family composition or work effort in response to the payments might lead to group differences in net family incomes that averaged either more or less than $3756.

Using income-to-needs as a measure of economic resources, the first row of Table 4 shows that the simple mean differences favor the high-cash group by 0.11 units in the pooled sample, which corresponds to a 11% increase in household income-to-needs. This effect is

**Table 1 | BFY baseline characteristics, by cash gift group**

| | Low-Cash Gift | High-Cash Gift | Std. Mean Difference | | |
|---|---|---|---|---|---|
| | Mean (SD) | Mean (SD) | Hedges' g | Cox's Index | *p*-value |
| Child | | | | | |
| Female | 0.50 | 0.48 | | −0.05 | 0.46 |
| Weight at birth (lb) | 7.13 (1.08) | 7.09 (1.01) | −0.04 | | 0.57 |
| Gestational age (weeks) | 39.09 (1.25) | 39.04 (1.24) | −0.04 | | 0.51 |
| Mother | | | | | |
| Age at birth (years) | 26.80 (5.82) | 27.38 (5.86) | 0.10 | | 0.11 |
| Education (years) | 11.88 (2.83) | 11.88 (2.96) | 0.00 | | 0.98 |
| Race/ethnicity | | | | | |
| White, non-Hispanic | 0.11 | 0.09 | | −0.13 | 0.13 |
| Black, non-Hispanic | 0.40 | 0.44 | | 0.10 | 0.09 |
| Multiple, non-Hispanic | 0.04 | 0.03 | | −0.18 | 0.37 |
| Other or unknown | 0.05 | 0.03 | | −0.32 | 0.07 |
| Hispanic | 0.41 | 0.41 | | 0.00 | 0.59 |
| Marital status | | | | | |
| Never married | 0.42 | 0.49 | | 0.17 | 0.02 |
| Single, living with partner | 0.26 | 0.22 | | −0.13 | 0.12 |
| Married | 0.21 | 0.21 | | 0.00 | 0.79 |
| Divorced/separated | 0.05 | 0.03 | | −0.32 | 0.06 |
| Other or unknown | 0.06 | 0.04 | | −0.26 | 0.40 |
| Health is good or better | 0.88 | 0.92 | | 0.27 | 0.04 |
| Depression (CES-D) | 0.68 (0.45) | 0.69 (0.46) | 0.02 | | 0.80 |
| Cigarettes per week during pregnancy | 5.05 (21.17) | 3.45 (11.76) | −0.09 | | 0.11 |
| Alcohol drinks per week during pregnancy | 0.17 (1.63) | 0.03 (0.39) | −0.11 | | 0.05 |
| Number of children born to mother | 2.40 (1.38) | 2.53 (1.41) | 0.09 | | 0.15 |
| Number of adults in household | 2.12 (1.00) | 2.03 (0.96) | −0.09 | | 0.16 |
| Biological father in household | 0.40 | 0.35 | | -0.13 | 0.15 |
| Household income ($1000 s) | 22.47 (21.36) | 20.92 (16.15) | −0.08 | | 0.22 |
| Household income unknown | 0.06 | 0.07 | | 0.10 | 0.48 |
| Household net worth ($1000 s) | −1.98 (28.64) | −3.31 (20.32) | −0.05 | | 0.42 |
| Household net worth unknown | 0.12 | 0.10 | | −0.12 | 0.64 |

Joint test: $\chi^2(29) = 33.98$, $p = 0.24$, $N = 1000$.

The *p* values were derived from a series of two-sided OLS bivariate regressions in which each respective baseline characteristic was regressed on the treatment status indicator using robust standard errors and site-level fixed effects. The joint test of orthogonality was conducted using a probit model with robust standard errors and site-level fixed effects. Standardized mean differences were calculated using Hedges' g for continuous variables and Cox's Index for dichotomous variables. The number of observations with non-missing baseline measures ranges between 531 and 600 for the low-cash gift group and between 358 and 400 for the high-cash gift group. If more than 10 cases were missing for a covariate, missing data dummies were included in the table and the joint test. If fewer than 10 cases were missing, missing data dummies were not included in the table but were included in the joint test. Chi-square tests of independence were conducted for the two categorical variables: mother race/ethnicity and mother marital status. For both tests, $p > 0.05$. All respondents with missing data on gestational age are in the control group, so this dummy is excluded from the joint test due to perfect collinearity. This results in a slightly smaller sample for the joint test. Joint test of orthogonality between treatment and baseline characteristics in the age 1, age 2, and age 3 follow-up samples yielded *p*-values of 0.39, 0.32, and 0.20, respectively.

*CES-D* Center for Epidemiologic Studies Depression Scale.

statistically significant after multiple testing adjustments. In terms of household income, the adjusted group difference amounts to $2850 for the pooled sample—a 0.13 standard deviation (SD) effect. Thus, the cash gift generated increases in economic resources and reductions in poverty over the first three years of the child's life. These difference, however, were modest, as more than half of the mothers receiving high-cash gifts were still residing in households with incomes less than the federal poverty threshold (data not shown in tables; see Gennetian et al.[43] for additional details about the impact of the cash gift on family expenditures, public benefit receipt, and parents' investments in children).

Did modest reduction in poverty and increases in income yield improvements in measures of economic hardship and pressure—food insecurity, non-food economic hardships, and worries about expenses? We had expected that higher cash gift payments would reduce economic hardship and pressure, but point estimates show no

evidence of reductions in these measures, with effect sizes ranging from +0.04 SD to +0.07 SD (Panel 2 in Table 4).

Results show that the impacts of the high-cash gift on measures of maternal well-being and psychological distress were not statistically significant. All the estimated impacts are in the opposite direction of what is predicted by the Family Stress Model (Panel 3, Table 5). Mothers in the high-cash gift group reported consistently higher levels of perceived stress, anxiety and parenting stress, but these coefficients were not statistically significant. It is also noteworthy that the high-cash gift impact appears to have increased maternal anxiety relative to the low-cash gift group as measured by the Beck Anxiety Inventory at age 1 (0.25 SD), but this impact fell to −0.01 at age 3.

Given unexpected negative significant impacts at age 1 on anxiety and negative (not statistically significant) point estimates for other aspects of maternal well-being, we conducted exploratory analyses to

**Table 2 | Summary of outcomes: economic resources and economic pressure**

| Family | Outcome | Hypoth. | Gift Group | Age 1 | | Age 2 | | Age 3 | | Pooled Sample (Ages 1–3) | |
|---|---|---|---|---|---|---|---|---|---|---|---|
| | | | | N | Mean (SD) | N | Mean (SD) | N | Mean (SD) | N | Mean (SD) |
| Panel 1: Economic Resources | | | | | | | | | | | |
| 1 | Income-to-needs ratio with gift | + | Low | | | 545 | 0.92 (0.84) | 542 | 1.04 (0.87) | 1087 | 0.98 (0.86) |
| | | | High | | | 377 | 0.99 (0.76) | 380 | 1.12 (0.86) | 757 | 1.05 (0.81) |
| 1 | Household income with gift ($1000 s, in 2019 dollars) | + | Low | | | 545 | 26.42 (24.95) | 542 | 29.78 (25.73) | 1087 | 28.10 (25.38) |
| | | | High | | | 377 | 28.85 (22.87) | 380 | 31.83 (25.26) | 757 | 30.35 (24.13) |
| Panel 2: Economic Pressure | | | | | | | | | | | |
| 2 | Food insecurity index | - | Low | 546 | 1.21 (1.67) | 544 | 1.16 (1.76) | 542 | 1.03 (1.77) | 1632 | 1.13 (1.73) |
| | | | High | 383 | 1.49 (1.77) | 377 | 1.20 (1.75) | 378 | 1.03 (1.67) | 1138 | 1.24 (1.74) |
| | Food insecurity index (5-item)[a] | - | Low | 546 | 1.21 (1.67) | 543 | 1.05 (1.55) | 541 | 0.93 (1.54) | 1630 | 1.07 (1.59) |
| | | | High | 383 | 1.49 (1.77) | 376 | 1.10 (1.54) | 378 | 0.96 (1.50) | 1137 | 1.18 (1.62) |
| 2 | Non-food economic hardship index | - | Low | 547 | 1.07 (1.14) | 544 | 0.96 (1.12) | 542 | 0.67 (0.91) | 1633 | 0.90 (1.08) |
| | | | High | 383 | 1.14 (1.21) | 377 | 1.05 (1.21) | 380 | 0.69 (0.88) | 1140 | 0.96 (1.13) |
| | Non-food economic hardship index (4-item)[a] | - | Low | 546 | 0.67 (0.86) | 543 | 0.64 (.83) | 542 | 0.67 (0.91) | 1631 | 0.66 (.87) |
| | | | High | 383 | 0.73 (0.90) | 377 | 0.70 (.91) | 380 | 0.69 (0.88) | 1140 | 0.71 (.89) |
| 2 | Expense worry | - | Low | 547 | 2.90 (1.65) | 544 | 2.61 (1.61) | 541 | 2.65 (1.63) | 1632 | 2.72 (1.63) |
| | | | High | 383 | 3.10 (1.59) | 375 | 2.70 (1.66) | 378 | 2.76 (1.56) | 1136 | 2.85 (1.61) |

We present a summary using all of the available items within each age (which match our impact analysis) and a summary using the subset items that appear in all ages (which make the measures more comparable across ages). We note the number of consistent items in parentheses. Outcomes are grouped into families following the conceptual model in Fig. 1. Preregistered hypothesized directions of the intervention effects are presented with "+" or "-" for a directional increase or decrease in the outcome, respectively. Household incomes across all years are inflation-adjusted to 2019 dollars, and the poverty line is based on the 2019 U.S. Census poverty threshold. Income-to-needs is the household income divided by the poverty line for a given family size and composition. Income and income-to-needs have been truncated at the 99th percentile.
[a]For five measures, the number of items that make up the index or scale are not identical across ages. See Supplementary Table 5 for details.

consider whether impacts on these scales were also found on subscales that comprise the full scales (Supplementary Table 6; not a pre-registered analysis). Specifically, we found that mothers in the high-cash gift group scored significantly higher on both the somatic and the psychological subindex of the Beck Anxiety Inventory. In addition, exploratory analyses indicated that the higher reports of parenting stress at age 1 in the high-cash gift group appeared to be driven by items in the parent aggravation subscale rather than items that assess parenting competence, whereas at age 2 both subscales were similarly elevated in the high-cash gift group.

With respect to mothers' reports of their romantic relationship quality (Panel 4 in Table 6), we find that the high-cash gifts produced an unexpectedly negative impact at age 3. In thinking about these results, it is important to remember that mothers only reported on their relationship quality in later waves of data collection if they were in a romantic relationship. Estimated impacts of the cash gifts on these outcomes might be biased if the cash gift also impacted whether mothers were in relationships. For this reason, it is important to note that there were no statistically significant differences between the high-cash and low-cash gift groups with respect to reporting of father involvement or being in a romantic relationship. This suggests that selection into relationships does not bias the intent-to-treat (ITT) estimates of these measures of relationship quality for these measures reported in Tables 4, 5, and 6.

The final set of outcomes that we consider are mother-reported and observed measures of parenting quality (Panel 5 in Table 6). In this group of outcomes, all estimated impacts are in the expected direction, although only one is statistically significant. Mothers in the high-cash gift group reported more frequent engagement in activities such as reading books or playing with their children than mothers in the low-cash gift group (0.16 SD, $p < 0.05$). The high-cash gift was not significantly associated with the quality of mothers' observed interactions with their children or mothers' reports of using spanking as a disciplinary strategy.

### Additional analysis and robustness checks

We conducted several robustness checks to determine whether our findings were sensitive to the estimation model specifications. First, we conducted an ITT analysis in which we used our preregistered family grouping, which differed slightly from the conceptual family grouping of variables derived from Fig. 1 (see Supplementary Table 7); Supplementary Table 8 provides impact estimates for the preregistered groupings, and results are similar to those presented in Tables 4, 5, and 6. The different sorting of measures into families adjusts $p$-values differently to correct for Type I error. Yet our findings are consistent with respect to the overall findings and are not sensitive to the different $p$-value adjustments.

To be sure that our results were not affected by differences in the survey administration, specifically the omission or inclusion of items for a scale across years, we estimated ITT impacts for the scales using only the common items (survey items included in all three ages). Results were consistent with those reported in Tables 4 and 5 (Supplementary Table 9).

Next, we used analytic weights that correct for imbalance of baseline characteristics across the high-cash and low-cash gift groups (Supplementary Table 10). In addition, we applied non-response weights based on all the covariates used in our regression models to adjust the pooled sample to reflect the characteristics of the full study sample at baseline (Supplementary Table 11). The pattern of these weighted results did not differ from those found in our covariate-adjusted regression models reported in Tables 4, 5, and 6. Findings based on datasets in which missing data had been imputed using chained equations were also similar to those reported in Tables 4, 5, and 6 (Supplementary Table 12). Taken together, these findings suggest that our results are not sensitive to the treatment of missing data.

Finally, our study sample is not large enough to detect modest differences in estimated impacts across subgroups. However, statistically significant differences in impacts for families with different characteristics did not emerge in exploratory analyses. We looked

**Table 3 | Summary of outcomes: parent psychological distress, interparental relationship quality, and parenting quality**

| Family | Outcome | Hypoth. | Gift Group | Age 1 N | Age 1 Mean (SD) | Age 2 N | Age 2 Mean (SD) | Age 3 N | Age 3 Mean (SD) | Pooled N | Pooled Mean (SD) |
|---|---|---|---|---|---|---|---|---|---|---|---|
| **Panel 3: Parent Psychological Distress** | | | | | | | | | | | |
| 3 | Perceived stress index | - | Low | 547 | 10.82 (6.35) | 543 | 10.32 (6.19) | 542 | 12.54 (7.41) | 1632 | 11.22 (6.73) |
| | | | High | 383 | 11.39 (6.04) | 377 | 10.73 (5.98) | 379 | 13.32 (6.69) | 1139 | 11.81 (6.34) |
| | Perceived stress index (9-item)[a] | - | Low | 547 | 10.82 (6.35) | 543 | 10.32 (6.19) | 542 | 10.91 (6.84) | 1632 | 10.68 (6.46) |
| | | | High | 383 | 11.39 (6.04) | 377 | 10.73 (5.98) | 379 | 11.59 (6.07) | 1139 | 11.24 (6.04) |
| 3 | Parenting stress index | - | Low | 547 | 15.05 (3.52) | 543 | 14.71 (3.55) | | | 1090 | 14.88 (3.54) |
| | | | High | 382 | 15.68 (3.42) | 375 | 15.18 (3.63) | | | 757 | 15.43 (3.53) |
| 3 | Maternal depression (PHQ-8) | - | Low | 547 | 3.72 (4.09) | 543 | 2.94 (3.91) | 541 | 3.42 (4.63) | 1631 | 3.36 (4.23) |
| | | | High | 383 | 3.91 (4.41) | 376 | 3.21 (4.26) | 378 | 3.23 (3.96) | 1137 | 3.45 (4.22) |
| 3 | Maternal anxiety (GAD-7) | - | Low | | | 543 | 2.49 (3.81) | 542 | 3.07 (4.36) | 1085 | 2.78 (4.10) |
| | | | High | | | 376 | 2.78 (4.14) | 379 | 3.06 (4.00) | 755 | 2.92 (4.07) |
| 3 | Maternal anxiety (Beck Anxiety Inventory)[a] | - | Low | 547 | 4.58 (6.57) | | | 542 | 5.26 (8.17) | 1089 | 4.92 (7.42) |
| | | | High | 383 | 5.94 (8.34) | | | 377 | 5.03 (7.42) | 760 | 5.49 (7.91) |
| 3 | Physiological stress (ln hair cortisol) | - | Low | 210 | 1.73 (1.37) | | | | | 210 | 1.73 (1.37) |
| | | | High | 154 | 1.89 (1.41) | | | | | 154 | 1.89 (1.41) |
| **Panel 4: Interparental Relationship Quality** | | | | | | | | | | | |
| 4 | Co-parenting relationship quality | + | Low | 429 | 19.36 (2.90) | 399 | 19.40 (2.77) | | | 828 | 19.38 (2.83) |
| | | | High | 291 | 18.95 (3.36) | 264 | 19.09 (3.13) | | | 555 | 19.02 (3.25) |
| 4 | Romantic Relationship Quality Index | + | Low | 325 | 26.98 (3.55) | 305 | 31.14 (2.68) | 466 | 30.21 (3.67) | 1096 | 29.51 (3.78) |
| | | | High | 247 | 26.58 (3.76) | 207 | 30.84 (2.95) | 327 | 29.60 (4.00) | 781 | 28.97 (4.04) |
| 4 | Romantic Relationship Quality Index (10-item)[a] | + | Low | 325 | 26.98 (3.55) | 305 | 28.16 (2.64) | 466 | 27.26 (3.58) | 1096 | 27.43 (3.37) |
| | | | High | 247 | 26.58 (3.76) | 207 | 27.88 (2.86) | 327 | 26.64 (3.92) | 781 | 26.95 (3.65) |
| 4 | Ever cut/bruised/seriously hurt by partner | - | Low | 325 | 0.08 (0.28) | 304 | 0.01 (.11) | | | 629 | 0.05 (.22) |
| | | | High | 247 | 0.07 (0.25) | 207 | 0.02 (0.15) | | | 454 | 0.05 (0.21) |
| 4 | Frequency of arguing | - | Low | 324 | 2.56 (1.02) | 305 | 2.36 (0.83) | | | 629 | 2.47 (0.94) |
| | | | High | 242 | 2.48 (0.96) | 207 | 2.48 (0.83) | | | 449 | 2.48 (0.90) |
| **Panel 5: Parenting Quality** | | | | | | | | | | | |
| 5 | Parent-Child Activities Index | + | Low | 547 | 10.29 (2.68) | 543 | 14.06 (2.98) | 537 | 12.68 (2.50) | 1627 | 12.33 (3.14) |
| | | | High | 382 | 10.78 (2.58) | 376 | 14.45 (2.83) | 378 | 13.07 (2.41) | 1136 | 12.76 (3.02) |
| 5 | Parent-Child Activities Index (3-item)[a] | + | Low | 547 | 8.82 (2.37) | 543 | 9.57 (2.13) | 537 | 9.44 (1.97) | 1627 | 9.27 (2.18) |
| | | | High | 382 | 9.27 (2.20) | 375 | 9.88 (1.85) | 377 | 9.78 (1.89) | 1134 | 9.64 (2.00) |
| 5 | Parent-child interaction (PICCOLO) | + | Low | 307 | 41.39 (5.48) | | | | | 307 | 41.39 (5.48) |
| | | | High | 236 | 41.63 (5.39) | | | | | 236 | 41.63 (5.39) |
| 5 | Spanking discipline strategy indicator | + | Low | 339 | 0.06 (0.24) | 542 | 0.19 (0.39) | 540 | 0.20 (0.40) | 1421 | 0.16 (0.37) |
| | | | High | 257 | 0.07 (0.25) | 372 | 0.14 (0.34) | 377 | 0.17 (0.38) | 1006 | 0.13 (0.34) |

[a]For five measures, the number of items that make up the index or scale are not identical across ages. See Supplementary Table 5 for details.

Notes: We present a summary using all of the available items within each age (which match our impact analysis) and a summary using the subset items that appear in all ages (which make the measures more comparable across ages). We note the number of consistent items in parentheses. Outcomes are grouped into families following the conceptual model in Fig. 1. Preregistered hypothesized directions of the intervention effects are presented with "+" or "-" for a directional increase or decrease in the outcome, respectively. PHQ-8=Personal Health Questionnaire Depression scale. GAD= Generalized Anxiety Disorder scale. PICCOLO=Parenting Interaction with Children: Checklist of Observations Linked to Outcomes.

**Table 4 | Summary of ITT estimates of the impacts of the BFY high-cash gift on family well-being and family processes: economic resources and economic pressure**

| Family | Outcome | Hypoth. | | Age 1 | Age 2 | Age 3 | Pooled Sample |
|---|---|---|---|---|---|---|---|
| Panel 1: Economic Resources | | | | | | | |
| 1 | Income-to-needs ratio (including the BFY gift) | + | Effect (con. interval) | | 0.10 (0.00, 0.20) | 0.11 (0.00, 0.22) | 0.11 (0.02, 0.19) |
| | | | Std. effect | | 0.12 | 0.13 | 0.13 |
| | | | N (deg. freedom) | | 922 (876) | 922 (876) | 1844 (956) |
| | | | p-value (WY p-value) | | 0.04 (0.06) | 0.04 (0.06) | 0.01 (0.02) |
| 1 | Household income with gift ($1000 s in 2019 dollars) | + | Effect (con. interval) | | 2.79 (−0.09, 5.68) | 2.76 (−0.32, 5.83) | 2.86 (0.37, 5.35) |
| | | | Std. effect | | 0.11 | 0.11 | 0.11 |
| | | | N (deg. freedom) | | 922 (876) | 922 (876) | 1844 (956) |
| | | | p-value (WY p-value) | | 0.06 (0.06) | 0.08 (0.08) | 0.03 (0.03) |
| Panel 2: Economic Pressure | | | | | | | |
| 2 | Food insecurity index | - | Effect (con. interval) | 0.23 (−0.00, 0.46) | −0.00 (−0.24, 0.23) | 0.05 (−0.17, 0.27) | 0.10 (−0.08, 0.28) |
| | | | Std. effect | 0.14 | −0.00 | 0.03 | 0.06 |
| | | | N (deg. freedom) | 929 (882) | 921 (875) | 920 (874) | 2770 (972) |
| | | | p-value (WY p-value) | 0.05 (0.14) | 0.98 (0.98) | 0.68 (0.88) | 0.27 (0.44) |
| 2 | Non-food economic hardship index | - | Effect (con. interval) | 0.04 (−0.12, 0.20) | 0.07 (−0.09, 0.23) | 0.02 (−0.09, 0.14) | 0.05 (−0.06, 0.16) |
| | | | Std. effect | 0.04 | 0.06 | 0.03 | 0.04 |
| | | | N (deg. freedom) | 930 (883) | 921 (875) | 922 (876) | 2773 (972) |
| | | | p-value (WY p-value) | 0.62 (0.62) | 0.38 (0.72) | 0.68 (0.88) | 0.41 (0.44) |
| 2 | Expense worry | - | Effect (con. interval) | 0.17 (−0.04, 0.38) | 0.08 (−0.14, 0.29) | 0.11 (−0.10, 0.31) | 0.12 (−0.04, 0.28) |
| | | | Std. effect | 0.10 | 0.05 | 0.06 | 0.07 |
| | | | N (deg. freedom) | 930 (883) | 919 (873) | 919 (873) | 2768 (972) |
| | | | p-value (WY p-value) | 0.12 (0.23) | 0.48 (0.72) | 0.31 (0.63) | 0.14 (0.34) |

Data collection occurred in July 2019 to June 2020 for age 1, July 2020 to July 2021 for age 2, and July 2021 to July 2022 for age 3. Each block of rows presents, for each outcome, the raw treatment effect with confidence intervals in parentheses; the standardized treatment effect size; number of observations with degrees of freedom in parentheses; and the p-value with Westfall-Young (WY) adjusted p-value in parentheses. The ITT estimates come from two-sided regressions with site fixed effects, controlling for baseline covariates, child age at interview, and phone interview status. Outcomes were standardized using the standard deviation of the low-cash gift within each age. We report the degrees of freedom computed as the sample size minus the number of parameters estimated in the model. This statistic is complicated in the pooled sample because we cluster the standard error to adjust for non-independence. For simplicity, we report the default degrees of freedom reported in most software, which is the number of clusters minus one. The p-value comes from analyses that do not correct for multiple outcomes, while the WY p-value is based on Westfall and Young[72] step-down resampling methods of addressing multiple hypothesis testing, where outcomes are grouped in families (following Fig. 1) and their p-values adjusted within each family. The "Pooled Sample" column presents estimates from analyses that pool observations across ages, adjust for age indicators, and cluster the standard error at the individual level. Preregistered, hypothesized directions of the intervention effects are presented with "+" or "-" for directional increase or decrease in the outcome, respectively. Household incomes across all years are inflation-adjusted to 2019 dollars, and the poverty line is based on the 2019 U.S. Census poverty threshold. Income-to-needs is the household income divided by the poverty line for a given family size and composition. Income and income-to-needs have been truncated at the 99th percentile.

specifically at differences between self-identified Black, non-Hispanic mothers and Hispanic mothers of any race because they are the two largest racial and ethnic groups in our sample. We did not find clear differences in the pattern of effects between Black and Hispanic mothers. We also compared impacts for mothers who were residing with the father of their child, compared to those who were not, and found that impacts were not significantly different. Finally, we split mothers by their reported income at baseline into a higher-income and lower-income group and compared impacts across these groups. Again, we found few substantive differences in these impacts. (Results of moderation analyses are in Supplementary Tables 13, 14 and 15). Future work will continue to explore possible differences across subgroups.

## Discussion

Considerable theory and empirical research have suggested that poverty and economic hardship negatively affects family processes and thus child and adolescent outcomes. Using data from a randomized controlled trial, we examined whether an unconditional monthly cash gift disbursed via debit cards to mothers with low income for ~36 months after their child's birth would generate causal impacts on economic hardship, maternal well-being and psychological distress, and mothers' relationships and parenting. We found that, when compared with a $20 monthly unconditional cash transfer, a monthly unconditional cash transfer of $333 increased both household income and reduced poverty by modest amounts (see also Gennetian et al.[43]). Contrary to expectations, this increase in income did not result in statistically significant reductions in economic hardship or worry. It also did not improve subjective well-being or psychological distress, nor did it improve mothers' romantic or co-parenting relationship quality. Indeed, there is some suggestion that the cash gift may have reduced the quality of romantic relationships and increased parenting stress. With a few select exceptions, the estimated impacts are largely consistent across all three ages of data collection, and our pooled estimates across all three waves have sufficient statistical power to detect small effects sizes of ~0.14.

**Table 5 | Summary of ITT estimates of the impacts of the BFY high-cash gift on family well-being and family processes: maternal psychological distress**

| Family | Outcome | Hypoth. | | Age 1 | Age 2 | Age 3 | Pooled Sample |
|---|---|---|---|---|---|---|---|
| Panel 3: Maternal Psychological Distress | | | | | | | |
| 3 | Perceived stress index | - | Effect (con. interval) | 0.62 (−0.18, 1.41) | 0.45 (−0.34, 1.24) | 0.75 (−0.14, 1.65) | 0.63 (−0.02, 10.27) |
| | | | Std. effect | 0.10 | 0.07 | 0.10 | 0.09 |
| | | | N (deg. freedom) | 930 (883) | 920 (874) | 921 (875) | 2771 (973) |
| | | | p-value (WY p-value) | 0.13 (0.31) | 0.26 (0.43) | 0.10 (0.27) | 0.06 (0.21) |
| 3 | Parenting stress index | - | Effect (con. interval) | 0.53 (0.06, 0.99) | 0.52 (0.06, 0.98) | | 0.52 (0.12, 0.92) |
| | | | Std. effect | 0.15 | 0.15 | | 0.15 |
| | | | N (deg. freedom) | 929 (882) | 918 (872) | | 1847 (964) |
| | | | p-value (WY p-value) | 0.03 (0.09) | 0.03 (0.08) | | 0.01 (0.06) |
| 3 | Maternal depression (PHQ-8) | - | Effect (con. interval) | 0.26 (−0.29, 0.80) | 0.33 (−0.20, 0.86) | −0.03 (−0.56, 0.50) | 0.19 (−0.22, 0.60) |
| | | | Std. effect | 0.06 | 0.08 | −0.01 | 0.05 |
| | | | N (deg. freedom) | 930 (883) | 919 (873) | 919 (873) | 2768 (973) |
| | | | p-value (WY p-value) | 0.35 (0.58) | 0.22 (0.43) | 0.91 (0.99) | 0.37 (0.60) |
| 3 | Maternal anxiety (GAD-7) | - | Effect (con. interval) | | 0.30 (−0.22, 0.82) | 0.17 (−0.34, 0.69) | 0.25 (−0.19, 0.68) |
| | | | Std. effect | | 0.08 | 0.04 | 0.06 |
| | | | N (deg. freedom) | | 919 (873) | 921 (875) | 1840 (956) |
| | | | p-value (WY p-value) | | 0.26 (0.43) | 0.51 (0.81) | 0.27 (0.55) |
| 3 | Maternal anxiety (Beck Anxiety Inventory) | - | Effect (con. interval) | 1.66 (0.66, 2.66) | | −0.04 (−1.02, 0.93) | 0.80 (−0.05, 1.64) |
| | | | Std. effect | 0.25 | | −0.01 | 0.12 |
| | | | N (deg. freedom) | 930 (883) | | 919 (873) | 1849 (967) |
| | | | p-value (WY p-value) | 0.00 (0.01) | | 0.93 (0.99) | 0.06 (0.21) |
| 3 | Physiological stress (ln hair cortisol) | - | Effect (con. interval) | 0.03 (−0.26, 0.32) | | | 0.03 (−0.26, 0.32) |
| | | | Std. effect | 0.02 | | | 0.02 |
| | | | N (deg. freedom) | 364 (317) | | | 364 (363) |
| | | | p-value (WY p-value) | 0.84 (0.84) | | | 0.84 (0.84) |

Data collection occurred in July 2019 to June 2020 for age 1, July 2020 to July 2021 for age 2, and July 2021 to July 2022 for age 3. Each block of rows presents, for each outcome, the raw treatment effect with confidence intervals in parentheses; the standardized treatment effect size; number of observations with degrees of freedom in parentheses; and the p-value with Westfall-Young (WY) adjusted p-value in parentheses. The ITT estimates come from two-sided regressions with site fixed effects, controlling for baseline covariates, child age at interview, and phone interview status. Outcomes were standardized using the standard deviation of the low-cash gift within each age. We report the degrees of freedom computed as the sample size minus the number of parameters estimated in the model. This statistic is complicated in the pooled sample because we cluster the standard error to adjust for non-independence. For simplicity, we report the default degrees of freedom reported in most software, which is the number of clusters minus one. The p-value comes from analyses that do not correct for multiple outcomes, while the WY p-value is based on Westfall and Young[72] step-down resampling methods of addressing multiple hypothesis testing, where outcomes are grouped in families (following Fig. 1) and their p-values adjusted within each family. The "Pooled Sample" column presents estimates from analyses that pool observations across ages, adjust for age indicators, and cluster the standard error at the individual level. Preregistered, hypothesized directions of the intervention effects are presented with "+" or "-" for directional increase or decrease in the outcome, respectively. PHQ-8=Personal Health Questionnaire Depression scale. GAD-7=General Anxiety Disorder-7.

Point estimates of impacts for all 13 outcome measures of maternal well-being and family processes were not only not statistically significant, but also contrary to directions predicted by the Family Stress Model. Figure 2 summarizes the effect size differences between the high-cash and low-cash gift groups and confidence intervals for the family-wise p-value adjustments from our pooled analysis across all three ages. This figure also shows whether impact estimates were (blue markers and lines) or were not (red markers and lines) in the expected direction. Figure 2 shows that impact estimates attained statistical significance only for the first (economic resources) and last (parenting quality) families of measures. Moreover, point estimates of impacts for all 13 components of the intervening family processes were contrary to directions predicted by the Family Stress Model.

Why might we have found so few improvements in material hardship, maternal well-being, and family processes of the magnitude we expected? One possibility is that $3996 of cash transfers each year for three years (the high-cash gift amount) was insufficient to lift families' incomes far enough above the poverty threshold to make a difference in their lives. The fact that the cash payments failed to reduce maternal reports of economic pressure and material hardship supports this hypothesis. The lack of statistically significant impacts on financial hardship may be due in part to the high costs of caregiving for young children and infants. Added expenses for diapers, clothes, and car seats, for example, add up and are not completely offset by increases in supports from public benefit programs (e.g., the U.S. Department of Agriculture's Special Supplemental Nutrition Program for Women, Infants, and Children). We also note that, initially, mothers

**Table 6 | Summary of ITT estimates of the impacts of the BFY high-cash gift on family well-being and family processes: interparental relationship quality and parenting quality**

| Family | Outcome | Hypoth. | | Age 1 | Age 2 | Age 3 | Pooled Sample |
|---|---|---|---|---|---|---|---|
| **Panel 4: Interparental Relationship Quality** | | | | | | | |
| 4 | Co-parenting relationship quality | + | Effect (con. interval) | −0.38 (−0.85, 0.09) | −0.34 (−0.80, 0.12) | | −0.34 (−0.73, 0.05) |
| | | | Std. effect | −0.13 | −0.12 | | −0.12 |
| | | | N (deg. freedom) | 720 (673) | 663 (617) | | 1383 (802) |
| | | | p-value (WY p-value) | 0.12 (0.36) | 0.15 (0.48) | | 0.09 (0.25) |
| 4 | Romantic Relationship Quality Index | + | Effect (con. interval) | −0.30 (−0.91, 0.32) | −0.32 (−0.81, 0.18) | −0.62 (−1.18, −0.06) | −0.47 (−0.86, −0.08) |
| | | | Std. effect | −0.08 | −0.12 | −0.17 | −0.14 |
| | | | N (deg. freedom) | 572 (525) | 512 (467) | 793 (747) | 1877 (900) |
| | | | p-value (WY p-value) | 0.34 (0.70) | 0.21 (0.48) | 0.03 (0.03) | 0.02 (0.08) |
| 4 | Ever cut/bruised/seriously hurt by partner | - | Effect (con. interval) | −0.02 (−0.07, 0.02) | 0.01 (−0.01, 0.04) | | −0.00 (−0.03, 0.02) |
| | | | Std. effect | −0.08 | 0.12 | | 0.02 |
| | | | N (deg. freedom) | 572 (525) | 511 (466) | | 1083 (770) |
| | | | p-value (WY p-value) | 0.35 (0.70) | 0.33 (0.48) | | 0.74 (0.75) |
| 4 | Frequency of arguing | - | Effect (con. interval) | −0.04 (−0.21, 0.14) | 0.11 (−0.04, 0.27) | | 0.05 (−0.07, 0.17) |
| | | | Std. effect | −0.04 | 0.13 | | 0.06 |
| | | | N (deg. freedom) | 566 (519) | 512 (467) | | 1078 (766) |
| | | | p-value (WY p-value) | 0.67 (0.70) | 0.15 (0.48) | | 0.43 (0.68) |
| **Panel 5: Parenting Quality** | | | | | | | |
| 5 | Parent-Child Activities Index | + | Effect (con. interval) | 0.44 (0.09, 0.79) | 0.43 (0.05, 0.81) | 0.38 (0.05, 0.72) | 0.42 (0.14, 0.70) |
| | | | Std. effect | 0.16 | 0.14 | 0.15 | 0.15 |
| | | | N (deg. freedom) | 929 (882) | 919 (873) | 915 (869) | 2763 (971) |
| | | | p-value (WY p-value) | 0.01 (0.04) | 0.03 (0.05) | 0.02 (0.05) | 0.00 (0.01) |
| 5 | Parent-child interaction (PICCOLO) | + | Effect (con. interval) | 0.53 (−0.42, 1.48) | | | 0.53 (−0.42, 1.48) |
| | | | Std. effect | 0.10 | | | 0.10 |
| | | | N (deg. freedom) | 543 (496) | | | 543 (542) |
| | | | p-value (WY p-value) | 0.28 (0.48) | | | 0.28 (0.28) |
| 5 | Spanking discipline strategy | + | Effect (con. interval) | 0.02 (−0.02, 0.06) | −0.05 (−0.10, −0.01) | −0.03 (−0.08, 0.02) | −0.03 (−0.06, 0.01) |
| | | | Std. effect | 0.08 | −0.14 | −0.07 | −0.06 |
| | | | N (deg. freedom) | 596 (549) | 914 (868) | 917 (871) | 2427 (959) |
| | | | p-value (WY p-value) | 0.40 (0.48) | 0.02 (0.05) | 0.27 (0.27) | 0.14 (0.27) |

Data collection occurred in July 2019 to June 2020 for age 1, July 2020 to July 2021 for age 2, and July 2021 to July 2022 for age 3. Each block of rows presents, for each outcome, the raw treatment effect with confidence intervals in parentheses; the standardized treatment effect size; number of observations with degrees of freedom in parentheses; and the p-value with Westfall-Young (WY) adjusted p-value in parentheses. The ITT estimates come from two-sided regressions with site fixed effects, controlling for baseline covariates, child age at interview, and phone interview status. Outcomes were standardized using the standard deviation of the low-cash gift within each age. We report the degrees of freedom computed as the sample size minus the number of parameters estimated in the model. This statistic is complicated in the pooled sample because we cluster the standard error to adjust for non-independence. For simplicity, we report the default degrees of freedom reported in most software, which is the number of clusters minus one. The p-value comes from analyses that do not correct for multiple outcomes, while the WY p-value is based on Westfall and Young[72] step-down resampling methods of addressing multiple hypothesis testing, where outcomes are grouped in families (following Fig. 1) and their p-values adjusted within each family. The "Pooled Sample" column presents estimates from analyses that pool observations across ages, adjust for age indicators, and cluster the standard error at the individual level. Preregistered, hypothesized directions of the intervention effects are presented with "+" or "-" for directional increase or decrease in the outcome, respectively. PICCOLO=Parenting Interaction with Children: Checklist of Observations Linked to Outcomes.

were told the payments would last for three years and four months, and the payments were not extended until their second year of gift receipt. It is possible that mothers' use of the cash gift might have been affected by the initially shorter time horizon. For example, mothers might have been hesitant to take on new expenses which might last for more than three years, such as a car loan or higher rent. They also may have been hesitant to decrease their income by cutting back on employment. Larger and longer-lasting increases in economic resources might improve key elements of the Family Stress Model.

An alternative explanation centers on the ages of the children. Samples in most other studies of cash transfers showing positive impacts on family processes consist of older children and adolescents. Family processes among families with young children and household income below the federal poverty threshold may be

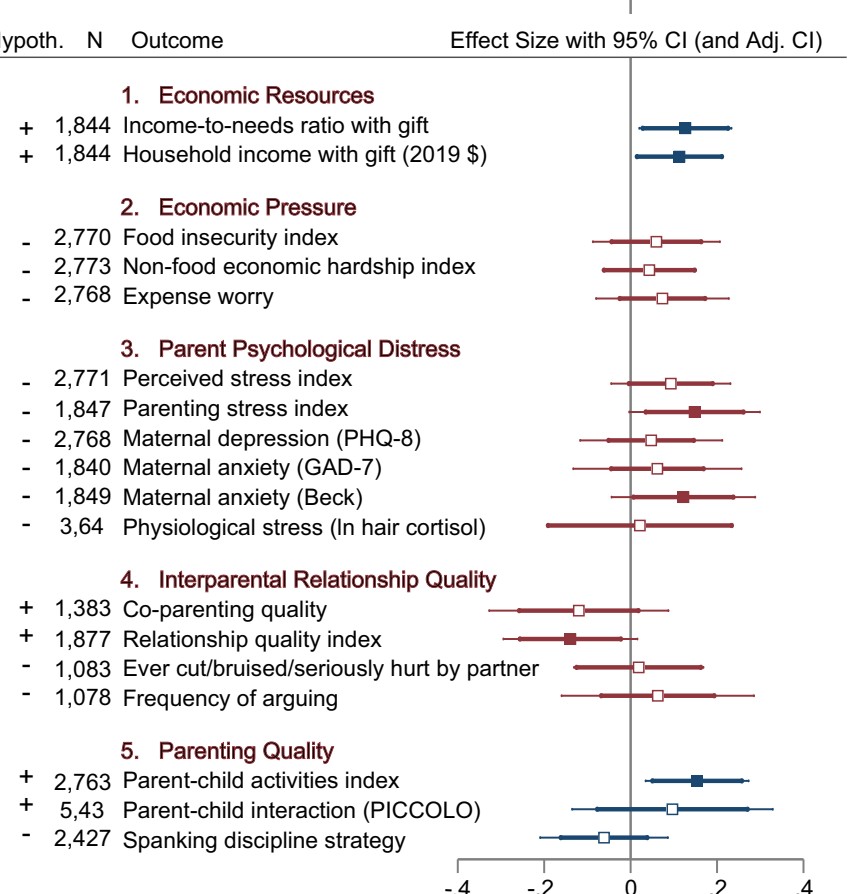

**Fig. 2 | Standardized effect size ITT estimates of the impact of the BFY high-cash gift with adjusted and unadjusted confidence intervals.** All presented estimates come from the two-sided intent-to-treat regression analysis pooling measures across ages found in Tables 4–6, clustering the standard errors at the individual level and controlling for baseline covariates, child age at interview, and phone interview status. Standardized treatment effects are represented by the square markers, and the whiskers (lines) represent unadjusted and adjusted 95% confidence interval (CI) estimates. Effect sizes are standardized by the standard deviation of the low-cash gift group and adjusted for multiple hypothesis testing with Westfall and Young[72] step-down resampling methods. Based on the adjusted p-value, degrees of freedom, and estimated effect size, the corresponding t-statistic and the standard error estimates were calculated to approximate the adjusted standard error and CI. Confidence intervals adjusted for multiple comparisons are represented by thin lines, and unadjusted confidence intervals are represented by thick lines. Filled square marker indicates that the estimate is statistically significant at the 0.05 level using the unadjusted p-value. Hypothesized direction of cash-gift treatment effects (preregistered) are presented in the "Hypoth." column with "+" indicating a directional increase in outcome and "-" indicating decrease in outcome. Blue shows impact estimates that were in the expected direction; red shows those that were not in the expected direction. Standardized estimates for spanking disciplinary strategy, a dichotomous outcome, come from a linear probability model. Raw linear probability model coefficients are presented in Table 4. Applying a logistic regression model and converting the resulting log of odds ratio into Cohen's d by a factor of the square root of 3 divided by pi estimates the effect sizes to be −0.14 for spanking disciplinary strategy (unadjusted p = 0.15).

affected differently by cash transfers than the family processes among families with older children and adolescents that are examined in other studies. Indeed, the birth of a child creates new expenses, such as diapers and baby food, and parents of infants often experience challenges in meeting caregiver obligations. Whatever the explanation, our study's surprising results indicate the need for further theoretical refinement and empirical testing of how cash transfers affect family processes, especially among families with young children.

It is also possible that mothers in the high-cash gift group experienced increased expectations or pressure to spend the cash in ways that benefited their children, and that this offset any positive benefits of the cash transfer. The Jaroszewicz et al.[37] study of cash payments during the pandemic found that being given some money (either $500 or $2000), but not enough to meet all their needs, may have made the gap between families' resources and needs more salient, and thus increased recipients' feelings of distress. Likewise, a study of microcredit lending in South Africa found that although men

experienced positive impacts on mental health, women did not, perhaps because they experience pressure to invest the money in a business which violated gender norms about their roles in the family[44]. Unfortunately, we do not have survey data that would directly shed light on these possibilities.

What might account for the decreases we see in maternal reports of romantic relationship quality at age 3? Given that this finding was unexpected, our possible explanations are post hoc and speculative. This finding is consistent with some theories that suggest that additional income will increases relationship conflict and violence for mothers who have male partners, either because the cash threatens the partner's status or because the partner will use threats or violence to extract income from the mother[11,13]. Nevertheless, prior studies of cash transfers have rarely shown negative impacts on romantic relationship quality or dimensions of intimate partner violence, so more work should be done to understand what aspects of the relationship are worse, and under what conditions such impacts might arise. It is noteworthy that this impact does not occur until the wave 3 survey

(around child age 3), and it will be instructive to see if these impacts persist at later ages.

If having the additional income—which is provided to the mother —introduces conflict into parental and romantic relationships, this might be another explanation for why the cash gift did not more broadly improve family well-being. This might be particularly true if the mothers feel less support and more criticism from their partners about their caregiving. It is worth noting, however, that other analyses of the BFY data do not show overall impacts on intimate partner violence[45].

Among our parenting measures, one out of three demonstrated a statistically significant impact in the hypothesized direction. Specifically, maternal self-reported engagement in learning activities, including reading and playing with their children, was higher among mothers in the high-cash gift group than among those in the low-cash gift group, while no statistically significant differences were found in maternal reports of frequency of spanking (a measure of harsh parenting) or the observed quality of parent-child interactions among the subsample who participated in a video-recorded ten-minute play session before the onset of the pandemic. However, it is important to recall that the parent-child observation was only collected for about 60% of the sample at age 1. The three indicators of parenting capture differing dimensions of parenting—it may be that time and engagement in activities with the child are affected more by income than the quality of interactions or discipline practices. Future work should measure multiple aspects of parenting to better tease apart income's possible differential effects.

The high-cash gift was associated with mothers reporting more time spent in learning activities with their child. Learning-related parent-child activities have been associated with positive child cognitive development as early as the first two years of life[46]. Indeed, Troller-Renfree et al.[47] found suggestive evidence that infants in the BFY high-cash gift group showed brain activity in a pattern that prior correlational studies have linked with the development of subsequent higher cognitive skills. It will be important to follow up with children at later ages to examine whether multiple years of the cash gifts affect child developmental outcomes.

The findings of this study should be interpreted in light of some limitations. First, this study used self-reported measures of most of the outcomes considered, including relationship quality and parenting stress, and self-reports may be differentially biased between the high-cash and low-cash gift groups. These measures have shown good psychometric properties in studies of populations with low income[48,49]. Yet, like many other studies using validated scales, our measures of key constructs did not demonstrate scalar or metric non-invariance across racial and ethnic or language groups. We did find configural, metric, and scalar invariance across our treatment and control groups. Nevertheless, future research should carefully consider how these constructs might differ across relevant groups and determine how best to measure these constructs in more comparable ways.

It is important to also note that null findings do not mean that the effects of the cash gift were zero. Our study had sufficient study design sensitivity to detect impacts across families of domains that were of a small to moderate sizes (see Fig. 2 and Tables 4, 5, and 6). Based on Bloom[50], for our pooled analysis, the minimum detectable effect size ranges from 0.14 to 0.33, with the median minimum detectable effect size of 0.22. As a result, it is possible that the payments had much smaller impacts that we could not detect. A challenge in this area of work is determining what size impact is meaningful, and this can be done by either relying on what is meaningful to individuals or based on cost-benefit analysis. Unfortunately, neither approach has been used to establish thresholds of the smallest effect size of interest for the outcomes in this paper. An important area for future work is to better understand what

magnitude of impacts will be of interest and reflect meaningful changes experienced by individuals.

A final important concern might be the that the COVID pandemic affected the validity of the study. A strength of the randomized controlled study is that unexpected events that affect both the treatment and comparison groups, such as the pandemic, do not affect the internal validity of the treatment impacts. That is, because both groups in this study experienced the same pandemic in the same communities, any differences observed between the groups can still be attributed to the cash gift. However, if the pandemic changed life in ways that made the additional cash more (or less) impactful, that might limit the external validity of the findings. For example, if the pandemic resulted in excessive economic hardship or anxiety, it might be that the cash gift had less impact because of the large amount of income needed to meet basic needs. One way to assess this is to consider whether the low-cash gift group experienced large shifts over time in their reports of the outcomes. Recall that each wave of data collection occurred over the course of a year, roughly July to June. The pandemic began in March 2020. Data from cell phone mobility and the imposition of stay-at-home orders suggest that all four of our research communities experienced the pandemic shutdown at the same time. This results in about 30% of the age 1 wave of data collection having been conducted during the pandemic. In contrast, we might consider most of the age 2 wave to have been completed during the pandemic, in that vaccines were not readily available to all adults until spring 2021.

During the age 2 data collection, we asked mothers about their experiences in the pandemic related to economic loss, health, and changes in their behavior. About 60% of mothers reported that they or someone in their household had lost income because of the pandemic. About 40% reported that they or someone in their household received unemployment insurance payments, and 74% reported that they received government stimulus payments. About 18% of mothers reported that they or someone in their household had been sick with COVID, and nearly 22% reported that a close friend or family member had died from a COVID infection. Finally, 75% of mothers reported that they had made major changes in their life because of the pandemic, and fully 94% of mothers reported that they engaged in social distancing.

Although the pandemic was a stressful experience for many families, its effect on the mothers in this study was likely heterogeneous. The expansions of safety net programs and provision of government stimulus payments may have resulted in some households having more money and resources than before the pandemic. It is noteworthy that the outcomes of low-cash gift group mothers are similar at each wave of data collection, and if anything, indicate minor improvements in economic stress over time. Moreover, an analysis by Premo et al.[51] of the BFY low-cash gift group data found that, if anything, mental health improved among mothers after the onset of the pandemic. This might suggest that pressures and routines for these mothers of young children were eased by stay-at-home orders. Thus, it is also noteworthy that mothers' mental health does not appear to worsen when children were two years of age, which corresponds to the peak of the pandemic (July 2020 to July 2021). Indeed, levels of anxiety and depression are overall low in this study and are lowest when children were two years old. Although this does not dismiss the concern that the families in our study and their experience of the cash gifts may have been affected by the COVID-19 pandemic, it does belie the simplistic explanation that families were so negatively affected by the pandemic that the cash gift did not matter.

In conclusion, our study findings suggest that providing $333 per month in unconditional cash support for about 36 months (of a planned 76 months) does not substantially improve subjective reports of economic pressure, reduce parent psychological distress, or improve relationship quality. Indeed, if anything, it may have worsened

the quality of romantic relationships and increased parenting stress. On the positive side, we found that such support resulted in improvements in the frequency of mothers' reports of engaging in stimulating activities with their young children. A full accounting of the benefits of the cash gifts will consider a broader set of economic, family, and child development outcomes[43,47,52,53]. As both the cash gifts and data collection are ongoing, future analyses will assess the extent to which monthly unconditional cash transfers affect family stress and well-being, as well as early childhood development, beyond the first three years of a child's life.

## Methods

We use data from Baby's First Years (BFY), an ongoing randomized controlled trial in which unconditional monthly cash transfers, hereafter referred to as "cash gifts," are being given to 1000 families. Between May 2018 and June 2019, people were invited to participate in the study shortly after giving birth, recruited from the postpartum wards of 12 U.S. hospitals in four metropolitan areas: New York City, New Orleans, Omaha, and the Twin Cities (Minneapolis and St. Paul). (The study materials did not ask about gender identity. We refer to these participants as mothers for ease of exposition.) When recruited into the study, all mothers reported an income below the federal poverty threshold. To participate, mothers had to be 18 or older; speak either English or Spanish; live in the state of recruitment with no immediate plans to move out of state; and report household income in the previous calendar year below the federal poverty threshold. In addition, they had to have a singleton pregnancy, their newborns must not have required intensive care, and newborns had to be discharged into the custody of their mothers. A total of 1000 mothers with newborns were enrolled in the study. The recruitment and subsequent three years of data collection were conducted by the Survey Research Center at the University of Michigan.

The institutional review boards of Teachers College (Protocol 18-210) and the New York State Psychiatric Institute (Protocol 7606) approved the study. All mothers provided written consent to participate in the study. To address ethical concerns regarding the possibility that cash gifts might coerce mothers to participate in research-based data collections, informed consent to participate in the research was uncoupled from the agreement to receive the monthly cash gift. Interviewers first described the longitudinal research study focused on child development and family life. After mothers consented to participate and were compensated for completing the baseline survey, the mothers were offered the opportunity to receive a monthly cash gift. Mothers who agreed to receive the cash gift were told the gift amount and their debit card was activated. Mothers were also informed that the study randomly assigned $333 or $20 monthly cash gifts.

At the time of recruitment, 400 mothers were randomly assigned to the group receiving $333 per month, and 600 mothers were randomly assigned to the group receiving $20 per month. Randomization occurred within each of the four sites. The first step in the randomization process was to create four rosters of 250 rows each, with 150 rows designated as "low-cash gifts" and 100 designated as "high-cash gifts." Each of the four 250-row rosters was then randomly ordered. Rows were assigned consecutively numbered cash gift IDs. As the 12-month recruitment period proceeded, it became clear that one site would not reach its goal of 250 recruited mothers, and this led to a roughly equal increase in the recruitment targets in the other three sites. To accomplish this, additional roster rows were created in each of these sites using the same randomization procedure. When aggregated, the 1000-row roster matched exactly the 40%/60% distribution of cash gifts across all possible respondents. A web-based application was used to access these rosters during the recruitment process, determine the high- vs. low-cash gift condition to be offered to each participant, record that the condition was offered, and communicate the gift value to the interviewer. Taken together, these procedures

ensured the integrity of the randomization process so that interviewers could not influence the assigned amount. As recruitment proceeded, three mothers who accepted the high-cash gift called within three days of its receipt to say they no longer wanted the debit card. These mothers were removed from the study, and three additional mothers were recruited, to ensure that the high-cash gift group had a sample size of 400 at baseline[54].

Following randomization, mothers were given a debit card that was activated at the hospital. Monthly cash gifts were loaded onto the card, which was branded "4MyBaby," on the evening prior to the day of the child's birthdate and accompanied by a text alert[55]. Efforts were made to ensure that, to the extent possible, the cash gift did not affect the mother's eligibility for safety net programs, such as the Supplemental Nutrition Assistance Program (SNAP), by working with state agencies and legislatures to make necessary rule changes. Mothers were initially told that the payments would continue for 40 months. In June 2021, when children were approaching their third birthdays, mothers were informed that cash gifts would continue for another year (for a total of 52 months). This was extended again in June 2022 for an additional two years (for a total of 76 months).

The first follow-up data were collected at approximately the time of the infants' first birthdays, between July 2019 and July 2020. This wave of follow-up data was originally collected during a home visit, which included an in-person maternal survey, a video recording of mother-child interactions, collection of a maternal hair sample for stress hormone (cortisol) analysis, and mobile electroencephalography (EEG) to measure the child's brain activity. Due to the COVID-19 pandemic, research staff switched from in-person to telephone-based data collection on March 13, 2020. At that point, it was no longer possible to collect video, hair, or EEG data. In all, 605 mothers completed data collection during a home visit, and 326 completed a survey by phone. Subsequent rounds of maternal surveys were administered by telephone at approximately the time of children's second and third birthdays.

After adjusting for a small number of mother-child separations, as well as infant and maternal deaths, the overall survey response rate was over 94% at each age (Supplementary Fig. 1). During the age 1 visit, videos of mother-child interactions were collected for 570 dyads (94% of the in-person sample) and hair samples from 409 mothers (68% of the in-person sample). The most common reasons for not completing the video-recorded interaction task included equipment malfunction, the child not being available, and mother's refusal. Reasons for not providing a hair sample include the use of corticosteroids and mother's refusal.

Preregistered hypotheses about measures and statistical procedures are available both from ClinicalTrials.gov (NCT03593356; first posted July 2018; https://clinicaltrials.gov/study/NCT03593356) and the American Economic Association registry (AEARCTR-0003262; first posted June 2019; https://www.socialscienceregistry.org/trials/3262). As the longitudinal study progressed, there were necessary deviations in the preregistered study plans. These deviations included, for example, changes in which measures were selected to be used, the number of items within indices, and at which ages some measures were collected. Many, but not all, of the deviations made from the original preregistration plan were made to adjust the study after the onset of the COVID-19 pandemic, which required us to postpone plans for in-person data collection from children's third birthdays to their fourth birthdays. All of the deviations in preregistration and can be found by consulting the ClinicalTrials.gov record history tab (https://clinicaltrials.gov/study/NCT03593356?term=NCT03593356&rank=1&a=1&tab=history&b=2#version-content-panel) or American Economic Association registry history documentation (https://www.socialscienceregistry.org/trials/3262/history). Both websites contain the same information.

In addition, this paper deviates in some ways from the pre-registered analysis plan. For example, we conduct additional analytic models to test the sensitivity of our findings to alternative specifications, and we estimate models that are pooled across all available ages of data. We identify throughout the paper which analytic models do and do not align with our preregistered plans.

Supplementary Table 4 provides a list of all primary and secondary preregistered outcomes through the age 3 data collection. The study is ongoing, and survey data for the first three years of the study are publicly available through ICPSR[56].

Following Fig. 1, we organized outcome measures into the following five categories: economic resources, economic pressure, maternal psychological distress, maternal co-parental and romantic relationship quality, and parenting quality. We provide more details on the available measures for each category in Supplementary Table 5. We report Cronbach's alpha for the scales and indices for the full sample and have conducted confirmatory factor analysis to ensure that there were no substantive differences in the factor structure of multi-item scales across racial and ethnic groups (Black vs. Hispanic) or by survey language (Spanish vs. English). However, as noted in the discussion section, while we confirmed configural measurement invariance, we did not find evidence of scalar or metric equivalence for scales across racial groups or by survey language. For some measures, there are some inconsistencies in the number of items included in the composite measures over three ages of data collection. This occurred both by design and because of errors in survey construction. Details for each composite measure are provided next.

### Economic resources
During all three follow-up surveys after baseline, mothers reported household pretax income in the previous calendar year and listed current adults and children in the household. We use data from ages 2 and 3, because our survey asked about income in the prior calendar year, and this ensures that information about income is from calendar years after random assignment (more information about income at age 1 can be found in Gennetian et al.[43]). We divided the average total household income, including the BFY cash gift, by the corresponding federal poverty threshold for a given household to create the "income-to-needs ratio." An income-to-needs ratio of 1.0 corresponds to 100% of the federal poverty threshold. In 2019, the federal poverty threshold for a family of four with two children was $2161 per month (or $25,926 for that year).

### Economic hardship
We have three measures of economic hardship generated from the maternal survey. First, we used an additive index of five or six items from the U.S. Department of Agriculture's short-form measure of food insecurity[57], which had high internal consistency ($\alpha = 0.85$–$0.87$) across ages. At age 1, one of the items, specifically about hunger, was inadvertently omitted, but the full scale was administered in the age 2 and age 3 surveys. Second, we created an additive index of four or five economic hardships (e.g., missing rent/mortgage payments) adapted from the economic stress index used in the Moving to Opportunity study[58]. This index of economically stressful events had low internal consistency ($\alpha = 0.46$–$0.54$). Modest internal consistency might be expected and acceptable for an index such as this in which the indicators are discrete events that are likely substitutes. In the age 3 survey, the item asking about missed phone payments was inadvertently omitted. Finally, we used a single item that was included in the economic stress index items (i.e., "worry about being able to meet monthly living expenses") with a six-point response scale, because it so closely aligns with economic worry.

### Mother's psychological distress
The maternal survey included four self-reported measures of mothers' stress and mental health. In addition, a sample of maternal hair was analyzed for cortisol concentration, providing a measure of physiological stress. General perceptions of life stress were measured by the Perceived Stress Scale[59,60], which had high internal consistency ($\alpha = 0.75$–$0.79$). During the age 1 and age 2 data collection waves, an item was inadvertently omitted, so the scale has nine rather than ten items; this item was added to the age 3 survey. It is notable that the mothers in this sample reported relatively lower levels of stress compared with national samples[61]. We also constructed a parenting stress index by summing two adapted indices: the parent aggravation index from the Panel Study of Income Dynamics' Child Development Supplement[62] and the parenting competence index created for the Getting Access to Income Now (GAIN) study (reverse coded; https://uwsc.wisc.edu/the-wisconsin-families-study-wiscfams/). Together, this parenting stress index had modest internal consistency ($\alpha = 0.55$). Depression was measured by the Personal Health Questionnaire Depression Scale[63] (PHQ-8), an additive index of eight items with high internal consistency ($\alpha = 0.84$). We use the scale as a continuous measure; note that across all three waves of data collection, only 9–10% of mothers scored above the suggested clinical cutoff, indicating moderate or higher levels of depression. This is comparable to rates of clinical depression found during the first postpartum year among mothers in the United States[64] but lower than expected for adults with income below 130% of the federal poverty threshold[65].

During the age 1 and age 3 waves of data collection, anxiety was measured by the Beck Anxiety Inventory[66], an additive scale of 21 items. In addition, anxiety was measured by the Generalized Anxiety Disorder (GAD-7) scale in the age 2 and age 3 data. Both measures of anxiety had high internal consistency across all waves ($\alpha = 0.90$–$0.92$). We use the GAD-7 scale as a continuous measure in our analysis. About 7–9% of mothers during the age 2 and age 3 data collection were above the suggested clinical cutoff indicating moderate or higher levels of anxiety. This is comparable to rates of elevated anxiety found in mothers during the antenatal period and first postpartum year[64,67].

We collected a hair sample from 409 of the 605 mothers (68%) who participated in the age 1 home interviews, but only 364 had usable values. Hair samples yield a measure of cortisol concentration in picograms per milligram (pg/mg). Values of 750 and higher ($n = 45$) are physiologically implausible and thus were not analyzed. Based on Lakens et al.[68], we adjusted two outlier values above 520 pg/mg by recoding them as 520 pg/mg. All values were then log transformed.

### Mothers' interparental and romantic partner relationship quality
In the age 1 and age 2 surveys, questions adapted from the Future of Families and Child Wellbeing Study[69] (FFCWS) were used to measure the quality of the mothers' co-parenting relationship in terms of support and trust. These questions were only asked if the father had spent time with their child in the last month. The additive index of seven items had high internal consistency ($\alpha = 0.90$).

The maternal survey also included questions about the quality of mothers' romantic relationships. Because these items were sensitive in nature, they were administered via audio computer-assisted self-interviewing (ACASI), which allows mothers to record their answers directly into a programmed computer. Because it was not possible to combine ACASI with telephone interviews during the pandemic, responses to these questions were not collected for the mothers surveyed by phone. If a mother reported that she was not currently in a romantic relationship during the age 1 survey, she was asked to report on the quality of the relationship with her most recent partner, such that some mothers reported on relationships that had ended. In later waves of data collection, these questions were only asked if mothers were in a current romantic relationship.

We constructed three measures of mother's romantic relationship quality. An indicator of domestic violence (whether the mother's partner ever cut, bruised, or seriously hurt her in a fight), and an item that describes how often the mother argues with the partner on important matters, both of which come from the FFCWS, were used as individual items. These items were only asked during the age 1 and age 2 surveys. Again, for the age 1 survey, this might have referred to a partner with whom the mother was no longer in a relationship. Finally, we measured the quality of the relationship between the mother and her romantic partner using a ten-item additive scale, also from FFCWS. The scale had good internal consistency ($\alpha = 0.83$).

**Parenting quality**

We measured three dimensions of parenting quality. First, to assess engagement in learning activities, we created an additive index of mothers' reported frequency of four or five activities that the mother engaged in with the child[70]. The items differed across waves because of the age appropriateness of activities. For example, at ages 2 and 3 (but not age 1), mothers reported how often they engaged in pretend play with their child. The activities index had adequate internal consistency ($\alpha = 0.61–0.67$). Second, at all ages, we assessed the use of harsh discipline through an indicator of whether the mother reported spanking her child in the past month because of misbehavior.

Finally, during the age 1 home visits, we assessed the quality of the parent-child interaction in a ten-minute video recording to capture affection, responsiveness, encouragement, and teaching in a total scale score using the Parenting Interactions with Children: Checklist of Observations Linked to Outcomes[71] (PICCOLO). We were able to record the interaction for 570 of the 605 mother-child dyads in the in-person, prepandemic sample (94%). The team of trained coders included a bilingual master coder and two additional master coders. In total, 135 out of the 540 videos, or 25%, were either double-coded or consensus-coded, and all intraclass correlation coefficient reliability values exceeded 0.75 as required[71]. After screening and processing the video for audio-video quality, we had usable data on parent-child interactions from 533 dyads.

**Control variables**

Data collected during a survey at the time of recruitment (prior to randomization) are used as covariates in our analysis. Items to be used as covariates were chosen because they are theoretically or empirically linked to the outcomes. These covariates included mother's age, mother's years of completed schooling, household income at baseline, net worth, general health, depressive symptoms, race and ethnicity, marital status, number of adults in the household, number of other children born to the mother, number of cigarettes smoked per week during pregnancy, number of alcoholic drinks consumed during pregnancy, biological father living with the mother, and the child's gender assigned at birth, birth weight, and gestational age at birth. We also included as covariates the age of the child in months and whether the age 1 interview was conducted in person or over the phone.

**Statistical power**

The overall sample size for the BFY study was designed at the start of the study such that, assuming 20% attrition by the time children were old enough to provide reliable data on our preregistered outcomes (age 36 months), an initial sample size of $n = 1000$ (and $n = 800$ at age 3), divided 40%/60% between the high- and low-cash gift groups, provided 80% statistical power to detect a 0.219 SD impact at age 3 at $p < 0.05$ in a two-tailed test. Our use of baseline covariates in impact estimation models was expected to reduce this minimal detectable effect size, while adjusting standard errors for sample clustering and multiple testing was expected to increase it. Based on exploratory analyses with data from the Future of Families study (a study with a demographically similar sample to BFY's), we expected that covariates

and clustering adjustments would roughly offset one another and would have little net impact on our power. Our preregistered use of the Westfall and Young[72] multiple testing adjustments was expected to increase the minimally detectable effect size, but the size of the increase depends on the number of measures in the family and their correlations.

When considering survey-based measures in our current analyses, the statistical power is slightly greater than in our original calculations, for two reasons. First, our attrition assumptions proved too pessimistic. As shown in Supplementary Fig. 1, all the response rates of data collection exceeded 92%. Repeating the original power analysis for $n = 920$, the detectable effect size for falls to 0.195 SD.

Second, because the same survey measures were gathered across multiple ages, we were able to pool age-specific samples and thus increase the effective sample size for our analysis. After adjusting for the nonindependence of the three age-specific samples, we calculated that our pooled analysis has 80% power to detect effects of about 0.14 SD with a two-sided test without any adjustments for multiple comparisons. The impact of the multiple adjustment to this effect size differs across outcome groupings (or families). Based on Bloom[50], the minimum detectable effect size for a specific analysis given a two-tailed test with $p < 0.05$ and 80% power can be computed by multiplying a given standard error by 2.8. Therefore, the minimum detectable effect size for each outcome in a family after adjustments for multiple outcomes can be calculated. This post hoc approach to study design sensitivity indicates that across the measures, our minimum detectable effect sizes range from 0.14 to 0.33, with the median of 0.22 across all 18 outcomes. A 0.22 effect size is a modest impact, reflecting an increase from the 50th percentile to the 59th percentile and from the 75th percentile to the 82nd percentile for a normally distributed scale score.

It is worth noting that the larger minimum detectable effect sizes reflect outcomes with smaller sample sizes. Hair cortisol and observations of parent-child interactions occurred in the age 1 data collection but were stopped with the onset of the COVID-19 pandemic. Given the much smaller sample sizes, power to detect small effects is much lower.

**Analytic strategy**

We used the random assignment design of the BFY clinical trial to estimate the causal effect of the $333-per-month cash gift payments on our outcome measures. ITT effects were estimated by regressing each dependent variable on the high-cash gift group indicator. Our preferred model pools all ages of data together because it increases the precision of our estimates. However, our preregistration plan did not include models that pooled outcomes across ages.

We adjusted all estimates for site indicators, as well as all covariates listed above, to increase the precision of our estimates and to account for any residual group differences in baseline characteristics following random assignment. We adjusted the standard errors using robust variance estimation techniques and clustered them at the participant level. We also estimate regressions for each age of data collection separately using the same specifications.

We addressed the possibility of false positives by estimating the statistical significance of the entire family (i.e., familywise error rate[73]) of outcomes using step-down resampling methods developed by Westfall and Young[72]. For the Westfall-Young correction, we place measures into preregistered conceptual families as specified in the model depicted in Fig. 1 and as detailed in Supplementary Table 5.

Unbiased ITT estimation requires that the high-cash and low-cash gift groups be similar on observed and unobserved characteristics. We assessed overall group balance using a probit model to jointly predict group assignment using site indicators and all the baseline characteristics listed in Table 1. At baseline and in our pooled analysis, we do not find systematic group differences ($p = 0.38$ and $p = 0.28$, respectively).

Nevertheless, at baseline and in the pooled sample, mothers in the high-cash gift group were more likely to have never been married and to have a racial and ethnic identification that was categorized as "other." At recruitment, mothers in the high-cash gift group were also more likely to report that their health was "good" or "excellent" (rather than "fair" or "poor"). To address potential bias from these minor differences and to increase the precision of ITT estimates, we control for the baseline characteristics shown in Table 1. This covariate-adjusted ITT model generates our preferred estimates of the high-cash gift's causal effects. Supplementary Tables 1 to 3 provide corresponding baseline balance information for our samples at each age of data collection.

We conducted several robustness checks to determine whether our findings were sensitive to the estimation model specifications. First, to better align with the Family Stress Model, some of the conceptual grouping of measures in the present analyses differed from those we preregistered (see Supplementary Table 7). Second, as noted, in some cases our outcome measures had differing numbers of survey items across ages due to error or by design. To be sure that our results were not affected by these differences in the survey administration, we estimated ITT impacts for the scales using only the common items (survey items included in all three surveys; this was not a preregistered analysis). Third, we used analytic weights that correct for imbalance of baseline characteristics across the high-cash and low-cash gift groups (Supplementary Table 10; this was not a preregistered analysis) to adjust the pooled sample to reflect the characteristics of the full study sample at baseline (Supplementary Table 11; this was not a preregistered analysis). These weights were constructed using a machine learning algorithm package called TWANG (Toolkit for Weighting and Analysis of Nonequivalent Groups)[74]. To confirm that results were not sensitive to attrition, we used multiple imputation by chained equation, or MICE, using linear regression and predictive mean matching and imputing 20 datasets (this was a preregistered analysis).

**Reporting summary**

Further information on research design is available in the Nature Portfolio Reporting Summary linked to this article.

## Data availability

The raw Baby's First Years data are publicly available at the Inter-university Consortium for Political and Social Research[56]. The publicly available data do not include one variable (child age) that is part of this analysis because sharing the variable would violate HIPAA data sharing rules. All other data are available.

## Code availability

The data cleaning and statistical analysis code used to create this paper is available at OPEN ICPSR[75]. All analysis code was written and conducted in Stata 18 and is available for download.

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

## Acknowledgements

Research reported in this publication was supported by the Eunice Kennedy Shriver National Institute of Child Health and Human Development of the National Institutes of Health under Award Number R01HD087384 (G.J.D., L.A.G., K.A.M., and K.G.N). The content is solely the responsibility of the authors and does not necessarily represent the official views of the National Institutes of Health. This research was additionally supported by the U.S. Department of Health and Human Services, Administration for Children and Families, Office of Planning, Research and Evaluation (R01HD087384; G.J.D., L.A.G., K.A.M., and K.G.N.); Office of Behavioral and Social Sciences Research-Office of the Director, National Institutes Of Health (R01HD087384; G.J.D., L.A.G., K.A.M. and K.G.N); Andrew and Julie Klingenstein Family Fund (K.G.N.); Annie E. Casey Foundation (214.0183; K.G.N.); Arrow Impact (K.G.N.); Bezos Family Foundation (K.G.N.); Bill and Melinda Gates Foundation (OPP1185312; K.G.N.); Bill Hammack and Janice Parmelee (K.A.M.); BCBS of Louisiana Foundation (K.A.M.); Brady Education Fund (G.J.D.); Chan Zuckerberg Initiative (Silicon Valley Community Foundation; 2017-177918; K.G.N.); Charles and Lynn Schusterman Family Philanthropies (13080; G.J.D. and K.G.N.); Child Welfare Fund (13-1624202; K.G.N.); Esther A. and Joseph Klingenstein Fund (K.G.N.); Ford Foundation (0170-0832; K.G.N.); Greater New Orleans Foundation (K.A.M.); Heising-Simons Foundation (542569; K.A.M.); Holland Foundation (542709; K.G.N.); Jacobs Foundation (102535: G.J.D.); JPB Foundation (1132, 2711, 3652; K.G.N.); J-PAL North America (S5341, 21-05679; G.J.D., and L.A.G.); Lozier Foundation (KN); New York City Mayor's Office for Economic Opportunity (CT1 069 20201415397; K.G.N.); Perigee Fund (K.G.N.); Robert Wood Johnson Foundation (71446, 75592, 78562; K.G.N.); Sherwood Foundation (4288; K.G.N.); Valhalla Foundation (K.G.N.); Weitz Family Foundation (K.G.N.); W.K. Kellogg Foundation (P3031579; K.G.N.); and three anonymous donors (G.J.D. and K.A.M.). The funders had no role in study design, data collection and analysis, decision to publish or preparation of the manuscript.

## Author contributions

In alphabetical order, G.J.D., N.A.F., L.A.G., K.A.M., K.G.N., and H.Y. equally contributed to the design and implementation of the study, project planning, data collection, data interpretation, and manuscript writing. S.H.-M. contributed to design and implementation of the qualitative sub-study, data interpretation, and manuscript writing. P.Y.Y. and S.H. conducted the analysis for this paper and contributed to the data interpretation and manuscript writing.

## Competing interests

The authors declare no competing interests.
