## [Transparent Peer Review file · Nature Communications]

Effects of Unconditional Cash Transfers on Family Processes and Wellbeing Among Mothers with Low Incomes

Corresponding Author: Dr Katherine Magnuson

Version 1:

Reviewer comments:

Reviewer #1

(Remarks to the Author)

Summary: This study describes the results of the RCT Baby's First Years, which provided unconditional \$333 monthly cash benefits to 600 new mothers compared to a control group of 400 mothers who received a smaller \$20 monthly benefit. The present study examines the effects on parental mental health and other related outcomes during an intermediate time point, showing mostly null results, with a few significant (and perhaps counterintuitive) findings. The manuscript addresses a critically important topic. It is very well written and methodologically sound with a substantial number of robustness checks. I have minor comments mostly related to the broader literature and the organization of several sections of the manuscript, as noted below.

- Review of CTC studies is not accurate. Both Batra and Kovski studies found improvements in mental health among parents, as did another study not cited (Nam et al JAMA Network Open 2024). Thus it might be better to report this literature as having mixed rather than null findings.
- The Methods section contains numerous instances where results are reported, e.g., demographic characteristics, summaries of health measures, etc. These should be moved to the Results.
- Similarly, some of the methods for the robustness checks are not described until the Results. They should first be introduced in the Methods section. And part of this section on Robustness Checks is more relevant for the Discussion, since it involves a commentary and contextualization of the results.
- Further context for the negative impacts on parenting stress and quality of romantic relationships may be drawn from the international literature, since the Discussion currently provides no citations on this topic. E.g., an RCT of small loans among low-income South Africans found that they improved depression for men but not women, perhaps due to differential gender norms (<https://link.springer.com/article/10.1186/1471-2458-8-409>). Other research finds similar possible gender-role-related conflicts with microcredit interventions: <https://www.sciencedirect.com/science/article/abs/pii/S0305750X9500124U>. While the present study is focused on cash distributions rather than loans, this prior literature (and the intersectionality literature) may nevertheless provide theoretical grounding for the observed effects in this study, i.e., the provision of cash to women leading to gendered conflicts with spouses, which could be incorporated into the Discussion and the theoretical model in Figure 1.
- For Appendix Figure 3, it would be helpful to have the dates of collection listed for each of the study waves, to help readers contextualize the data collection within the events of the pandemic.

Reviewer #2

(Remarks to the Author)

I congratulate the authors on a well-conceived and well-executed randomized trial examining the effects on maternal wellbeing and child development of a substantial cash transfer program. Quasi-experimental studies as well as earlier lesser-powered intervention studies have suggested strong potential effects of cash transfers, and there has been widespread policy enthusiasm for expanding such transfers. This "Baby's First Years" study is the strongest yet to carefully test early childhood cash transfer effects in the U.S., making this manuscript a good fit for a top general interest journal.

Methodologically I find the study to be excellent. Hypotheses were pre-registered; outcomes are appropriate and well-measured; the study was well-powered for meaningful effect sizes; analyses properly account for multiple testing; randomization led to reasonably balanced arms (with results not sensitive to controls for baseline imbalances); attrition was

modest and fairly balanced by arm, etc. I do have a few specific suggestions for improving the manuscript:

1. Please add a short summary of the results in Gennetian (2022), which are highly relevant to interpreting the results in the present paper. E.g., the lack of effects on parental time allocation such as work hours, no effect on child care spending, and the highly diffuse spending patterns suggesting that only a small portion of funds were spent on child-focused items (such as books) directly related to the outcomes studied here. I suggest also returning to these Gennetian results in the Discussion when interpreting the small effect sizes found here.
2. It would be helpful to include a more expansive literature review of the effect of cash transfers on relationship quality (e.g., related to first and seventh paragraphs of the Discussion). In the literature from low and middle income countries there have been heterogeneous effects, e.g. with Hidrobo and Fernald (Journal of Health Economics 2013) reporting that cash transfers could have adverse effects on emotional violence when the woman's education was greater than her partner's, and Bobonis et al. (American Economic Journal: Economic Policy, 2013) presenting an intrahousehold model consistent with cash increasing violent threats.
3. Please note in the manuscript that parents were informed at enrollment that they would be receiving the transfers for 40 (?) months, and note when they were subsequently told that it would be extended, as this is relevant for understanding their behavioral responses.
4. Please clarify the relationship between the pre-registration at ClinicalTrials.gov versus the AEA RCT registry. It appears to me that the latter is more complete, in which case it should be cited as well.
5. To facilitate reader understanding and to promote research reproducibility, further details would be helpful on the exact questions analyzed. I suggest an appendix showing the wording of each of the questions included in each index (the AEA RCT registry has some of this information, but the final versions used should be presented here), along with ideally a link to the code used to create the analytic variables. (This would also help clarify e.g. whether family composition adjustments in the poverty line variable use endogenous time-varying current family composition or instead the pre-program baseline family composition.)
6. Please further clarify in the table and figure notes which analyses controlled for baseline characteristics. Relatedly, the terminology "adjusted and unadjusted" is confusing when referring to confidence intervals as opposed to the more standard use of the terms related to covariate adjustment. Perhaps refer instead to "confidence intervals both uncorrected and corrected for multiple testing."
7. Timepoint terminology is also confusing, sometimes using "waves 1-3," sometimes "12, 24, and 36 months," and sometimes "age 1, 2, and 3." I suggest using a single terminology (personally I find "months 12, 24, and 36" to be least confusing).
8. I found the Table 4 discussion and interpretation (just before the Robustness Checks section) difficult to follow. (Please elaborate and clarify the writing here. Note also that Panel 1 of table 4 swaps the order of the outcomes compared to other tables and "ES" in the column heading is not defined.)
9. There are typos in the Appendix Table 5A notes. Line 2 appears to be cutoff, and the column numbering is incorrect.

Reviewer #3

(Remarks to the Author)

Widening inequalities have led to growing concerns about the impact of poverty on children's development – and parenting is generally theorized to play an important mediating role in the impact of poverty on the child. This study adopts a simple experimental design to test whether reducing financial stress through unconditional cash transfers (UCT) might lead to improvements in the quality of parents' interactions with their infants and toddlers. The evidence from low- and middle-income countries shows modest effects on child and parental wellbeing, with mixed findings from the few UCT studies within the USA.

The results suggest that moderate amounts of monthly cash transfers have little beneficial impact upon parents' self-reported economic hardship, interparental relationship quality, or wellbeing (despite modestly improving economic resources). Likewise, the authors report no significant group differences in the quality of observed mother-child interactions, although those receiving higher cash transfers did report engaging in enriching activities with their young child more often.

Describing this sample is obviously a complex and challenging task, and this is one area in which the authors could add more relevant information. Table 1 offers data on a comprehensive list of variables, but there is little reporting of how these coincide. For example, how many mothers in the study sample were in a lone-parent household with more than one child? This is a group that is likely to be especially affected by financial strain, and for whom the UCT is unlikely to go as far as for mothers with more than one child. That is, UCT may only bear fruit if the sums involved are likely to make a substantial difference to household finances. Perhaps this is covered in another paper from this study, but a brief recap would be welcome.

Effect sizes are clearly presented and support the authors' measured conclusions, but several methodological issues limit the interpretation of the study results findings and deserve note. First and most importantly, while the authors did conduct CFAs to show similarity of factor structure across languages and racial and ethnic groups, they didn't test for metric and scalar measurement invariance, leaving the equivalence of scores across different demographic groups open to question.

Second, the low-income mothers in the sample reported relatively low levels of stress and depression, which raises questions about the validity of the psychological distress scales. Understanding these results would be easier if the authors included the full range of scores / max possible score for the outcome measures.

Third, the internal consistency of measures of economic hardship and parenting stress is described as 'modest' – this is perhaps misleading, as some of the Cronbach alpha values are very low, and so this should be acknowledged as a limitation. It's good to see transparency re human error in survey construction, but there does appear to be a rather high degree of inconsistency between time-points in which survey items were included – for future studies, it might be worth including a comment about the factors that may have led to errors not being spotted in time (e.g., the shift to remote working?).

Turning to the interpretation of the results, it would be helpful for the reader to have a richer discussion of why high UCTs should be associated with more time spent on enriching activities but unrelated to the observed quality of parent-child interactions. One plausible explanation is that there are legacy effects of poverty that are not easily remediated by UCT. Alternatively, it may simply be that there's an informant effect – participants were not asked to report on the quality of their interactions with their infant, so these two measures (quant and qual) are assessed in different ways...

Alongside the discussion of the model with the pooled sample, the main discussion section should also cover: (i) wave-specific findings (e.g., maternal anxiety at age 1, romantic relationship at age 3); (ii) the likely impact of the pandemic on study findings. Here it is worth noting that the results from this study may well have been compromised by factors beyond the researchers' control – namely the multiple stresses experienced by families with young children during the COVID-19 pandemic; and (iii) moderating effects of child age (what factors might explain the weaker results in families with infants / toddlers than in families of older children?)

Minor points / typos:

- The title in the pdf has "An Experimental Analysis" written twice, once with a '?'
- Abstract should specify that monthly cash transfers were provided for 3 years.
- Use en dashes (–) to show ranges in tables and text; hyphens (-) look like a negative sign and make confidence intervals harder to read.
- Unclear what 'efforts' were made to ensure that cash transfers did not affect mothers' eligibility for safety net programs.
- p.11 fourth line from the bottom – "these difference"
- p. 15 "poverty and economic hardship negatively affects"
- p. 24 "estimate is statistically significant results"

Reviewer #4

(Remarks to the Author)

Review notes by Zoltan Kekecs:

The manuscript describes a large scale multi-site study (the Baby's First Years study) assessing the effects of an unconditional cash gift of 333 USD per month to low-income mothers of newborns on factors thought to affect the child development. This longitudinal study is a huge undertaking, and our field needs more of these rigorous experimental studies deployed on the populations of interest. I especially commend the authors for the great job at retention of participants over multiple years even through the pandemic, and for the gargantuan work of transparently documenting this research in the trial registry and keeping it up-to-date despite the huge number of outcomes assessed in this study. The paper is also well written. As a methods expert in my review I focused on the methodological and especially the analytical aspects of the study. I hope that my suggestions detailed below can help to facilitate the highest quality methodological reporting this magnificent study deserves:

- There were some deviations from the originally preregistered protocol and outcomes measures (both primary and secondary) according to ClinicalTrials.gov. The authors should be transparent about these changes. I understand that in a complex study like this it is very hard to provide a comprehensive picture of all changes in a paper with limited space, so I suggest simply writing that there were deviations, give a brief summary of the types of things that were changed and are relevant to the current paper, and direct readers to check ClinicalTrials.gov registry version history for the details.
- In a study with a substantial unconditional cash gift for one of the groups I imagine that participants are motivated to get into the treatment group rather than the control group. I suggest that the authors present statistics to the reader about the success of the planned randomization procedure. One such information is already presented by the authors: the equivalence of the groups on different baseline characteristics. However, it would be great to see more information about the randomization procedure and its success. Although the authors say that details of the randomization is provided in Noble (2021), that publication barely has any information on the procedure. How did the authors ensure that the randomization process was not biased, or that group allocation was concealed? Also, what was the acceptance rate of the group allocation?
- For a project of such magnitude, I find the sample size justification and the power analysis surprisingly underwhelming. The proposed models to test the hypotheses are pretty complex, so the most adequate power analysis method would have

been simulation-based power analysis. Also, the described analyses in the power analysis section are not reproducible, because not enough details are shared. I suggest sharing the statistical analysis code to make power analysis reproducible.

- The pooling of the data from subsequent years of data collection is not part of the preregistered analysis plan. I agree that this is a sensible move, but this should be marked as a deviation from the registered plan.
- The authors say that McGuire et al. (2022) finds significant average impact ($d=.13$) for studies that boost income by about 20% (the same size of income increase that is in our study). However, they later report in the Results section that the gain in the treatment group corresponds to an 11% increase in household income-to-needs, so I find it a bit of an exaggeration that we can expect that $d = 0.14$ effect size can be detected with 80% power by this study. Especially that the power analysis was not done on the intended analysis. I suggest re-estimating the sensitivity of the study using simulation-based power analysis, where the different adjustments and covariates can be factored in.
- In the Results section the authors talk about un-adjusted and adjusted significance levels, and “marginal significance”. Since the authors planned to do adjustment for multiple testing, they should only report the adjusted test results. I found the current reporting confusing, in some cases where it is not stated in the text whether the finding is for an adjusted or un-adjusted test I found myself wondering whether this result was significant after adjustment. Also, in null hypothesis significance testing, there is no such thing as “marginally significant”, so I suggest dropping this phrase. If a finding does not cross the significance threshold after adjustment for multiple testing, it is simply not significant.
- I am not sure what MDES refers to. It should be explained in text. I suspect it is minimally detectable effect size.
- The authors say “To explore whether estimated ITT impacts are equivalent to 0, we conducted two one-sided tests using our regression framework (Lakens, Scheel & Isager, 2018), in which the MDES was set to .20 and .14 (approximately the smallest effect size detectable in our single wave and pooled models respectively).” These equivalence tests are usually conducted against a smallest effect size of interest (SESOI) threshold, not against the minimally detectable effect size. In an underpowered study, it is easily possible that an intervention would turn out to have an effect equivalent to zero according to the minimally detectable effect, but this would not be the case against the smallest effect of relevance/interest. And it could be true the other way around in a highly powered study. So I suggest setting up SESOI thresholds for each outcome based on what effect would still be meaningful for the population of interest.
- Please give further detail about how the equivalence tests were conducted, so that it is easy to reproduce. I suggest sharing analysis code and data, so that analytical reproducibility can be tested, and the research community can benefit from the hard work of putting together this complex analysis pipeline.
- I would not use formal equivalent testing in this study, because it was not pre-registered. I would simply interpret the effect estimates and dispersion statistics as exploratory findings in comparison to SESOI bounds set by the authors. That does not require formal statistical testing, and the results should be interpreted cautiously as exploratory findings that should be confirmed in a confirmatory study.
- The modeling approach seems a bit outdated. I understand that the authors would like to stick to the preregistered analysis approach, but it might be good to conduct a linear mixed effect model as a robustness check with at least family ID and state of residence as random effect predictors. (I would like to again emphasize the importance of sharing analysis code and data within the paper.)

Reviewer #5

(Remarks to the Author)

Version 2:

Reviewer comments:

Reviewer #1

(Remarks to the Author)

The authors have responded to all reviewer critiques thoroughly and satisfactorily.

(Remarks on code availability)

Reviewer #2

(Remarks to the Author)

Thank you for the comprehensive response and revision, I have no further comments.

(Remarks on code availability)

Reviewer #3

(Remarks to the Author)

The authors have done a good job of responding to my comments and I'm now happy to support publication.

(Remarks on code availability)

Reviewer #4

(Remarks to the Author)

I am grateful to the authors for their diligent work in addressing my comments and the comments of the other referees. I understand that this must have taken quite some time and effort, but it paid off. The manuscript is now significantly improved, and all of my previous comments have been addressed to my satisfaction.

The analysis code attached to the submission is very thorough and well documented. I especially like that it has a very detailed readme file that documents the contents of all code segments and provides instructions on how to replicate their results. (I need to say that I don't use stata, so I am unable to do a computational reproducibility check, but based on the thorough documentation, I have confidence that that would check out as well.)

One remaining minor concern is that the link to the analysis code included in the Code Availability section leads to a repository with multiple folders apparently containing supplementary data for different publications, and I was not able to figure out which one contains the relevant code for this particular paper. It would be important for the link to lead to the relevant file or folder immediately, to make sure that the reader can access the analysis code for this particular study.

Since the authors revised some aspects of the manuscript related to statistical power, the editor asked me to pay special attention to issues related to power, so below are my detailed assessment of the sections related to this issue:

Overall I would say that the author's approach and revised discussion section is reasonable. I find it quite transparent, and the conclusions they draw seem to be accurate and not exaggerated. Based on reading the paper, looking at the reported results and the thoroughness of the report and analysis I largely agree with the conclusions and the wording of the discussion.

In this revised version the authors are quite transparent in that their preregistration is public, and also in their reporting of the sample size rationale. In the previous version of the manuscript they provided a post-hoc justification for the sample size and the power, which built on results that came out after their initial power analysis and sample size target was set. In this post-hoc analysis they refer to McGuire et al 2022 who found that studies which provide an increase of 20% income usually result in an effect size of $d = 0.13$ on average. Adding that since their own study also used a 20% income increase, their minimally detectable effect is in the right ballpark. I criticized this statement, since the income increase was later empirically measured by the authors at 11%. To address my criticism, the authors took this statement out and revised it to be more accurate, and they no longer use this statement to imply that their study was adequately powered.

When assessing the manuscript as a whole I find it transparent, that the intervention the authors have applied might be expected to have a smaller effect than what the study was capable of easily detecting. I find the statements of the authors that are currently in the revised manuscript and its discussion section to be accurate and not misleading.

There is one notable exception: "...and our pooled estimates have sufficient statistical power to detect small effects sizes of approximately .14.", first of all, it would be better to use the word "sensitivity" instead of statistical power in this sentence, because power is determined pre-measurement, and once the data has been collected, we usually refer to post-hoc calculated "power" as sensitivity of the design instead. This slight modification in wording could be applied to the relevant sentences throughout the manuscript. This is a minor thing that statisticians would pick up but that will probably not make a difference for 99% of the readers.

A bigger issue with this sentence is the number .14, which is the lowest number they found in their analysis, but for some of the outcomes this number was as high as .33. I would find it more reasonable to use the median in this section as an approximate sensitivity, which was .22 as reported in the Statistical power section.

As I mentioned, the manuscript in its entirety is sufficiently transparent for my taste. When looking at the discussion section on its own, I still find it reasonable with the revised limitations included in the latest version of the manuscript. Maybe the abstract could be revised to reflect the deficiency in the range of detectable effects. For example in the abstract it could be added to the end of the conclusions that "... but the possibility of very small effects are not ruled out by our results".

One thing I noticed during the re-reading is that the numbers related to sensitivity mentioned in the discussion don't quite match up with the numbers reported in the statistical power section: Statistical power section: "This post-hoc approach to statistical power indicates that across the measures, our minimum detectable effect sizes range from .14 to .33, with the median of .22 across all 18 outcomes." Discussion section: "Based on Bloom (1995) for our pooled analysis, the minimum detectable effect size ranges from .14 to .30, with the average of minimum detectable effect size of .23." This should be sorted out for the numbers (and the metrics mentioned: median vs. mean) to be consistent.

Overall impression:

I am not a domain expert, so I might not have a full picture here, but based on the manuscript and preregistration, I find this study and the manuscript to be very credible. I was convinced by the results and the cited other findings from the literature that this intervention (at least implemented in the way the authors did) is unlikely to yield substantial benefits to people. Yes, very small effects are not ruled out, but these effects, even if they might exist, would seem negligible to me (a domain expert might have a different read on this).

(Remarks on code availability)
See comments to authors above.

Reviewer #5

(Remarks to the Author)

(Remarks on code availability)

Response to REVIEWER COMMENTS

Reviewer #1 (Remarks to the Author):

Summary: This study describes the results of the RCT Baby's First Years, which provided unconditional \$333 monthly cash benefits to 600 new mothers compared to a control group of 400 mothers who received a smaller \$20 monthly benefit. The present study examines the effects on parental mental health and other related outcomes during an intermediate time point, showing mostly null results, with a few significant (and perhaps counterintuitive) findings. The manuscript addresses a critically important topic. It is very well written and methodologically sound with a substantial number of robustness checks. I have minor comments mostly related to the broader literature and the organization of several sections of the manuscript, as noted below.

- Review of CTC studies is not accurate. Both Batra and Kovski studies found improvements in mental health among parents, as did another study not cited (Nam et al JAMA Network Open 2024). Thus it might be better to report this literature as having mixed rather than null findings.

We now describe the CTC studies as providing mixed results (page 4) and include the Nam & Kwon reference (which was published after this paper was originally submitted).

- The Methods section contains numerous instances where results are reported, e.g., demographic characteristics, summaries of health measures, etc. These should be moved to the Results.

We have moved the description of our sample to the results section.

- Similarly, some of the methods for the robustness checks are not described until the Results. They should first be introduced in the Methods section. And part of this section on Robustness Checks is more relevant for the Discussion, since it involves a commentary and contextualization of the results.

We have moved the description the robustness checks in the Analytic Strategy section of the paper and moved the contextualization of results to the Discussion.

- Further context for the negative impacts on parenting stress and quality of romantic relationships may be drawn from the international literature, since the Discussion currently provides no citations on this topic. E.g., an RCT of small loans among low-income South Africans found that they improved depression for men but not women, perhaps due to differential gender norms

(<https://link.springer.com/article/10.1186/1471-2458-8-409>). Other research finds similar possible gender-role-related conflicts with microcredit interventions:

<https://www.sciencedirect.com/science/article/abs/pii/S0305750X9500124U>. While the present study is focused on cash distributions rather than loans, this prior literature (and the intersectionality literature) may nevertheless provide theoretical grounding for the observed effects in this study, i.e., the provision of cash to women leading to gendered conflicts with spouses, which could be incorporated into the Discussion and the theoretical model in Figure 1.

We have added to the discussion some mention of prior studies suggesting that microcredit/cash transfers might be less beneficial for women in terms of their mental health or relationship quality because of gender norms. We also note that another reviewer requested that we do not interpret impacts that are marginally significant, as a result we do not interpret the parenting stress impact as significant (it is $p < .10$) and only the age-3 romantic relationship impact as negative (rather than the pooled impacts which is also $p < .10$).

- For Appendix Figure 3, it would be helpful to have the dates of collection listed for each of the study waves, to help readers contextualize the data collection within the events of the pandemic.

We added the following the Note to Appendix Table 3 “Age 1 data was collected in June of 2019 to June of 2020; Age 2 Data was collected July of 2020 to June of 2021; Age 3 Data was collected June of 2021 to June of 2022” and also added these dates to Appendix Figure 1.

Reviewer #2 (Remarks to the Author):

I congratulate the authors on a well-conceived and well-executed randomized trial examining the effects on maternal wellbeing and child development of a substantial cash transfer program. Quasi-experimental studies as well as earlier lesser-powered intervention studies have suggested strong potential effects of cash transfers, and there has been widespread policy enthusiasm for expanding such transfers. This “Baby’s First Years” study is the strongest yet to carefully test early childhood cash transfer effects in the U.S., making this manuscript a good fit for a top general interest journal.

Methodologically I find the study to be excellent. Hypotheses were pre-registered; outcomes are appropriate and well-measured; the study was well-powered for meaningful effect sizes; analyses properly account for multiple testing; randomization led to reasonably balanced arms (with results not sensitive to controls for baseline imbalances); attrition was modest and fairly balanced by arm, etc. I do have a few specific suggestions for improving the manuscript:

1. Please add a short summary of the results in Gennetian (2022), which are highly relevant to interpreting the results in the present paper. E.g., the lack of effects on parental time allocation such as work hours, no effect on child care spending, and the highly diffuse spending patterns suggesting that only a small portion of funds were spent on child-focused items (such as books) directly related to the outcomes studied here. I suggest also returning to these Gennetian results in the Discussion when interpreting the small effect sizes found here.

There is an updated version of this work (Gennetian et al 2024) and the paper’s findings has been briefly summarized in the introduction as suggested. We also return to this in the discussion section.

2. It would be helpful to include a more expansive literature review of the effect of cash transfers on relationship quality (e.g., related to first and seventh paragraphs of the Discussion). In the literature from low and middle income countries there have been heterogenous effects, e.g. with Hidrobo and Fernald (Journal of Health Economics 2013) reporting that cash transfers could have adverse effects on emotional violence when the woman’s education was greater than her partner’s, and Bobonis et al. (American Economic Journal: Economic Policy, 2013) presenting an intrahousehold model consistent with cash increasing violent threats.

We added this point in the introduction section, suggesting that gendered resource theory and household bargaining theory offer differing predictions than the Family Stress Model. We also mention Hidrobo & Fernald’s findings in the introduction. Finally, we have also edited the discussion section to revisit these theories as a possible explanation for the findings.

3. Please note in the manuscript that parents were informed at enrollment that they would be receiving the transfers for 40 (?) months, and note when they were subsequently told that it would be extended, as this is relevant for understanding their behavioral responses.

This information has been added to the manuscript when we describe the cash gift, and we return to it in the discussion to point out that the initially shorter time horizon might have affected their spending choices and behavioral responses.

4. Please clarify the relationship between the pre-registration at ClinicalTrials.gov versus the AEA RCT registry. It appears to me that the latter is more complete, in which case it should be cited as well. Both registries include the same exact scientific information, although the presentation of the information differs. We now include both registries in the text so readers can look at them both. (We use the AEA RCT registry because it is preferred by economic journals).

5. To facilitate reader understanding and to promote research reproducibility, further details would be helpful on the exact questions analyzed. I suggest an appendix showing the wording of each of the questions included in each index (the AEA RCT registry has some of this information, but the final versions used should be presented here), along with ideally a link to the code used to create the analytic variables. (This would also help clarify e.g. whether family composition adjustments in the poverty line variable use endogenous time-varying current family composition or instead the pre-program baseline family composition.)

We provide this in a new table—Appendix Table 3—which provides all the wording/description of the variables. To answer your specific question, we use the family size at each age to calculate income-to-needs. As a result, this measure would capture any changes in income as well as any changes in “family”—capturing both forms of behavioral responses.

We also have provided all the code for this paper as a supplementary file with the submission (See the “ReadMe” text for an explanation of which files are used in each analysis. Also note that we have added `_STATA` to the end of the dofile names so these can be easily identified for the review process). These files will be made publicly available when the paper is published (we have deposited code for our prior papers on OpenICPSR; Magnuson, Katherine, Noble, Kimberly, Duncan, Greg, Fox, Nathan, Gennetian, Lisa, Yoshikawa, Hirokazu, and Halpern-Meekin, Sarah. Baby’s First Years Supplemental Files. Ann Arbor, MI: Inter-university Consortium for Political and Social Research [distributor], 2024-06-20. <https://doi.org/10.3886/E159422V3>).

6. Please further clarify in the table and figure notes which analyses controlled for baseline characteristics. Relatedly, the terminology “adjusted and unadjusted” is confusing when referring to confidence intervals as opposed to the more standard use of the terms related to covariate adjustment. Perhaps refer instead to “confidence intervals both uncorrected and corrected for multiple testing.”

We have implemented these suggestions in the table notes. And revised the tables to provide the “p-value | WY p-value” which we hope clears up any confusion about terminology related to which p-values are corrected for multiple outcomes.

7. Timepoint terminology is also confusing, sometimes using “waves 1-3,” sometimes “12, 24, and 36 months,” and sometimes “age 1, 2, and 3.” I suggest using a single terminology (personally I find “months 12, 24, and 36” to be least confusing).

We have decided to use age 1,2,3 consistently through-out the paper and tables, and have edited to be consistent.

8. I found the Table 4 discussion and interpretation (just before the Robustness Checks section) difficult to follow. (Please elaborate and clarify the writing here. Note also that Panel 1 of table 4 swaps the order of the outcomes compared to other tables and “ES” in the column heading is not defined.)

This analysis/table was added at the request of the Editor because of the journal’s policy that all null findings must include an equivalence test. However, based on the suggestion of another reviewer who

notes that this equivalence test was not part of our pre-registration plan, we have removed this table from the paper (see below).

9. There are typos in the Appendix Table 5A notes. Line 2 appears to be cutoff, and the column numbering is incorrect.

These typos and mistakes have been corrected.

Reviewer #3 (Remarks to the Author):

Widening inequalities have led to growing concerns about the impact of poverty on children's development – and parenting is generally theorized to play an important mediating role in the impact of poverty on the child. This study adopts a simple experimental design to test whether reducing financial stress through unconditional cash transfers (UCT) might lead to improvements in the quality of parents' interactions with their infants and toddlers. The evidence from low- and middle-income countries shows modest effects on child and parental wellbeing, with mixed findings from the few UCT studies within the USA.

The results suggest that moderate amounts of monthly cash transfers have little beneficial impact upon parents' self-reported economic hardship, interparental relationship quality, or wellbeing (despite modestly improving economic resources). Likewise, the authors report no significant group differences in the quality of observed mother-child interactions, although those receiving higher cash transfers did report engaging in enriching activities with their young child more often.

- Describing this sample is obviously a complex and challenging task, and this is one area in which the authors could add more relevant information. Table 1 offers data on a comprehensive list of variables, but there is little reporting of how these coincide. For example, how many mothers in the study sample were in a lone-parent household with more than one child? This is a group that is likely to be especially affected by financial strain, and for whom the UCT is unlikely to go as far as for mothers with more than one child. That is, UCT may only bear fruit if the sums involved are likely to make a substantial difference to household finances. Perhaps this is covered in another paper from this study, but a brief recap would be welcome.

We have revised the text to report that 24% of mothers lived in households with no other adults and two or more children. In designing the study, we opted to enroll a sample of participants that were heterogeneous and reflected the broad population of low-income mothers. As a result of this decision and the overall study sample size, we do not have enough statistical power to investigate impacts for subgroups that are relatively small (i.e. single mothers with multiple children).

- Effect sizes are clearly presented and support the authors' measured conclusions, but several methodological issues limit the interpretation of the study results findings and deserve note. First and most importantly, while the authors did conduct CFAs to show similarity of factor structure across languages and racial and ethnic groups, they didn't test for metric and scalar measurement invariance, leaving the equivalence of scores across different demographic groups open to question.

We tested our measures for configural, metric and scalar measurement invariance for Black and Hispanic mothers and for those who took the survey in Spanish (vs English). We found evidence of configural equivalence, but not metric or scalar invariance with just a few (rare) exceptions. We have revised the manuscript to note this and raise as a limitation in the discussion.

What this means for our findings, however, is not clear. As more and more scales are found to have measurement non-invariance the field is still grappling with what that means from a conceptual and practical level (see Welzel, C., Brunkert, L., Kruse, S., & Inglehart, R. F. , 2023 ; Meuleman, B., Żóttak, T., Pokropek, A., Davidov, E., Muthén, B., Oberski, D. L., ... & Schmidt, P., 2023; Robitzsch, A., & Lüdtke, O., 2023). The concern about measurement non-invariance is fundamental to research in cross-cultural psychology, and it seems many analyses suggest that metric non-invariance is common in this work.

Most of the concern with measurement invariance centers on the point that it is ill-advised to make comparisons across groups where there is known metric non-invariance. In our case, the invariance is found across demographic and language groups—but the focus of our RCT is on one pair of groups -- the high and low cash gift groups and we found full configural, metric and scalar invariance for our outcomes across treatment groups (using the Chi-square tests and CFI differences of less than .01 suggested by Putnick & Bornstein, 2016).

We selected scales that had been either widely used in prior research with diverse samples or had been used in a study of a similar population. This strategy is obviously not ideal for finding measures that are fully comparable across populations that differ by race, ethnicity or language. That said, if we held ourselves to the standard of requiring metric measurement invariance we would not be able to find measures of depression, anxiety, etc. that meet that standard. We believe the best way to handle this is to inform readers and mention this as a limitation of the study.

Robitzsch, A., & Lüdtke, O. (2023). Why full, partial, or approximate measurement invariance are not a prerequisite for meaningful and valid group comparisons. *Structural Equation Modeling: A Multidisciplinary Journal*, 30(6), 859-870.

Meuleman, B., Żóttak, T., Pokropek, A., Davidov, E., Muthén, B., Oberski, D. L., ... & Schmidt, P. (2023). Why measurement invariance is important in comparative research. A response to Welzel et al.(2021). *Sociological methods & research*, 52(3), 1401-1419.

Putnick, D. L., & Bornstein, M. H. (2016). Measurement invariance conventions and reporting: The state of the art and future directions for psychological research. *Developmental review*, 41, 71-90.

Welzel, C., Brunkert, L., Kruse, S., & Inglehart, R. F. (2023). Non-invariance? An overstated problem with misconceived causes. *Sociological Methods & Research*, 52(3), 1368-1400.

- Second, the low-income mothers in the sample reported relatively low levels of stress and depression, which raises questions about the validity of the psychological distress scales. Understanding these results would be easier if the authors included the full range of scores / max possible score for the outcome measures.

The range of scores for all outcomes has been added to Appendix Table 3, which we have revised to provide the wording of all items.

- Third, the internal consistency of measures of economic hardship and parenting stress is described as 'modest' – this is perhaps misleading, as some of the Cronbach alpha values are very low, and so this should be acknowledged as a limitation.

We have revised our discussion to include this point, as well as the point about metric and scalar non-invariance in our measures. We also note that economic hardship is intended to be an index rather than a scale – it is a summary of multiple indicators of hardship that do not all represent the same underlying phenomenon or latent dimension, but rather when summed provide a general summary of a family's economic circumstances. Put another way we do not necessarily expect high inter-item correlations for these items. For example, given limited money, a family may either choose to miss a rent payment or

miss a utility payment. With respect to parenting stress, we note that there are only 7 items overall and two subconstructs within this scale; as a result, it does have a low alpha (which is a function of inter-item correlations and the number of items—it is noteworthy that the subscales have higher alphas). Still, we appreciate that we should not refer to these items having modest internal consistency, so we have revised the text. As noted before, we have revised the text in the conclusion to raise measurement as a limitation.

-It's good to see transparency re human error in survey construction, but there does appear to be a rather high degree of inconsistency between time-points in which survey items were included – for future studies, it might be worth including a comment about the factors that may have led to errors not being spotted in time (e.g., the shift to remote working?).

The inconsistency of our measures were driven both by errors and by conceptual/measurement decisions. In this paper, three inconsistencies were driven by errors (Economic Hardship, Food insecurity, and perceived stress—in each of these scales one item was omitted during a wave). The other inconsistencies had to do with conceptual changes in measurement (i.e., parent child activities are different with infants compared with toddlers) or time scales (asking about last calendar year income meant that we didn't have consistent year incomes during age 1 data collection). This is a complex project and survey, and we worked closely with the University of Michigan's Survey Research Center to implement the survey instrument in their proprietary software and conduct the field interviews. They have great expertise in this type of work, and unfortunately, we cannot pinpoint where in the process things went wrong for these three items and therefore, we do not speculate about causes in the manuscript.

- Turning to the interpretation of the results, it would be helpful for the reader to have a richer discussion of why high UCTs should be associated with more time spent on enriching activities but unrelated to the observed quality of parent-child interactions. One plausible explanation is that there are legacy effects of poverty that are not easily remediated by UCT. Alternatively, it may simply be that there's an informant effect – participants were not asked to report on the quality of their interactions with their infant, so these two measures (quant and qual) are assessed in different ways...

These represent differing dimensions of parenting, and are assessed in differing way, and we have revised the discussion to reflect this. Mother's reports of activities reflect the time parents spend engaged with their child and the quality of interaction is scored by a trained rater based on the video of a structure activity.

- Alongside the discussion of the model with the pooled sample, the main discussion section should also cover: (i) wave-specific findings (e.g., maternal anxiety at age 1, romantic relationship at age 3); (ii) the likely impact of the pandemic on study findings. Here it is worth noting that the results from this study may well have been compromised by factors beyond the researchers' control – namely the multiple stresses experienced by families with young children during the COVID-19 pandemic; and (iii) moderating effects of child age (what factors might explain the weaker results in families with infants / toddlers than in families of older children?)

We have revised the discussion to include discussion of both wave specific results. As requested by another reviewer, we moved our discussion of COVID from the results to the discussion section. Finally, we discuss the children's young ages as an explanation for our findings.

Minor points / typos:

- The title in the pdf has "An Experimental Analysis" written twice, once with a '?'

- Abstract should specify that monthly cash transfers were provided for 3 years.
 - Use en dashes (–) to show ranges in tables and text; hyphens (-) look like a negative sign and make confidence intervals harder to read.
 - Unclear what ‘efforts’ were made to ensure that cash transfers did not affect mothers’ eligibility for safety net programs.
 - p.11 fourth line from the bottom – “these difference”
 - p. 15 “poverty and economic hardship negatively affects”
 - p. 24 “estimate is statistically significant results”
- We have corrected all of these typos.

Reviewer #4 (Remarks to the Author):

Review notes by Zoltan Kekecs:

The manuscript describes a large scale multi-site study (the Baby’s First Years study) assessing the effects of an unconditional cash gift of 333 USD per month to low-income mothers of newborns on factors thought to affect the child development. This longitudinal study is a huge undertaking, and our field needs more of these rigorous experimental studies deployed on the populations of interest. I especially commend the authors for the great job at retention of participants over multiple years even through the pandemic, and for the gargantuan work of transparently documenting this research in the trial registry and keeping it up-to-date despite the huge number of outcomes assessed in this study. The paper is also well written. As a methods expert in my review I focused on the methodological and especially the analytical aspects of the study. I hope that my suggestions detailed below can help to facilitate the highest quality methodological reporting this magnificent study deserves:

- There were some deviations from the originally preregistered protocol and outcomes measures (both primary and secondary) according to ClinicalTrials.gov. The authors should be transparent about these changes. I understand that in a complex study like this it is very hard to provide a comprehensive picture of all changes in a paper with limited space, so I suggest simply writing that there were deviations, give a brief summary of the types of things that were changed and are relevant to the current paper, and direct readers to check ClinicalTrials.gov registry version history for the details.

We have implemented this suggestion and now also include information for the AEA pre-registration which includes the same information. The duplicate AEA pre-registration was undertaken because some of the economics journals we were hoping to publish some of our papers in strongly urged it.

- In a study with a substantial unconditional cash gift for one of the groups I imagine that participants are motivated to get into the treatment group rather than the control group. I suggest that the authors present statistics to the reader about the success of the planned randomization procedure. One such information is already presented by the authors: the equivalence of the groups on different baseline characteristics. However, it would be great to see more information about the randomization procedure and its success. Although the authors say that details of the randomization is provided in Noble (2021), that publication barely has any information on the procedure. How did the authors ensure that the randomization process was not biased, or that group allocation was concealed? Also, what was the acceptance rate of the group allocation?

We agree that the Noble (2021) reference does not provide sufficient detail on randomization and have substituted a reference to the study’s website (<https://www.babysfirstyears.com/data-and-documentation>), which provides a full accounting. We have also added substantially more text explaining the details of the randomization procedure, which were developed and implemented in collaboration with our survey partner (the University of Michigan’s Survey Research Center).

- For a project of such magnitude, I find the sample size justification and the power analysis surprisingly underwhelming. The proposed models to test the hypotheses are pretty complex, so the most adequate power analysis method would have been simulation-based power analysis. Also, the described analyses in the power analysis section are not reproducible, because not enough details are shared. I suggest sharing the statistical analysis code to make power analysis reproducible.

We appreciate the observation that our power analysis is underwhelming. In fact, in the paper we provide the power analysis that we conducted in 2016 for the funding application for the project and included in our original pre-registration. Of course, at that time we had collected no data.

We began with a simple calculation of sample size based on a target minimum detectable effect size (MDES) ($\sim .20$ SD), power level, and sample allocation between high and low cash gift groups. We then considered that the most important adjustments to this naïve calculation would be for i) efficiency gains owing to controls for baseline covariates; ii) efficiency losses owing to clustering the sample by site and iii) multiple testing adjustments. To determine the importance of i) and ii), we found an existing study of newborns born into a clustered national sample of hospitals in large urban areas and followed into adolescence that provided child outcomes (our primary hypotheses) at age 3. Relying on Bloom's (1995) method for converting standard errors into MDEs, we first estimated a regression of standardized test scores on a randomly-generated dichotomous variable with a 60/40 split using a randomly selected $n=800$ cases from the data set. Confirming Bloom's conclusions, the standard error on the dichotomous measure, when multiplied by the 2.8 constant converting a standard error into a MDES for the two-tailed, 80% power assumption in our power analysis reproduced the $\sim .20$ SD effect size MDES.

We were then in a position to estimate the extent to which the standard error changes in the presence of the kinds of baseline covariates that would be available in our planned data collection and after adjusting for clustering at the urban area level. Neither adjustment changed the standard error appreciably, leading us to conclude that the simple calculation would serve as a reliable guide to needed sample size.

As we considered how to respond to this comment, we realized that while the simulation-based power analysis might be useful, that in some sense the die is cast. We designed the study and selected the sample based on the power analysis we describe. Of course, at this point, the actual data and the sample size is no longer expected but rather known. And it turns out that standard errors in the collected data are very much in line with what we expected based on our original calculations.

Nature requires reporting confidence intervals rather than SEs so we thought that rather than do a micro simulation, we should revise this text to explain where we were referring to a priori statistical power and refer the reader to Figure 2 and the Tables to consider the confidence intervals and implied minimum detectable effect sizes for the current analysis. We have revised the text, and now we leverage Bloom (1995) to show that our minimum detectable effect sizes for our pooled analysis range from .14 to .30 for measures that we have in all three years (and on average are .23). We provide more detail about this in our text. See our response to your comments further below for more detail on the specific MDEs.

Bloom, H. S. (1995). Minimum detectable effects: A simple way to report the statistical power of experimental designs. *Evaluation review*, 19(5), 547-556.

-The pooling of the data from subsequent years of data collection is not part of the preregistered analysis plan. I agree that this is a sensible move, but this should be marked as a deviation from the registered plan.

We have added text to explain that the pooled approach is a deviation from pre-registered plans. We also have revised the text to note which analyses were and were not pre-registered in the text of the manuscript.

- The authors say that McGuire et al. (2022) finds significant average impact ($d=.13$) for studies that boost income by about 20% (the same size of income increase that is in our study). However, they later report in the Results section that the gift in the treatment group corresponds to an 11% increase in household income-to-needs, so I find it a bit of an exaggeration that we can expect that $d = 0.14$ effect size can be detected with 80% power by this study. Especially that the power analysis was not done on the intended analysis. I suggest re-estimating the sensitivity of the study using simulation-based power analysis, where the different adjustments and covariates can be factored in.

The 20% figure (and our assertion that our cash transfer was similar) is based on the amount of the transfer relative to baseline family income. This is the most appropriate denominator because family incomes after randomization and the receipt of the case payments could be affected by behavioral responses (decreases in earnings or other forms of income as a result of cash transfer). In fact, there were non-significant reductions in mothers' and other household members' earnings in the high cash gift group post-randomization, which accounts for why the impact on income was an increase of 11% rather than closer to 20% (as reported in Gennetian et al. 2024).

But, we see your concern with the power analysis, and we have removed the text that suggests we should be able to detect a .14 effect size.

The minimum detectable effect size (MDES) is easy to compute by multiplying the resulting standard error for the standardized coefficient by 2.8 for two tailed, $P < .05$, and 80% statistical power (Bloom, 1995). Standard errors are not reported in the manuscript because of Nature's preference for confidence intervals, but we have estimated MDESes using Bloom's (1995) rule based on Westfall Young adjusted SEs and this suggests the following MDES : income-to-needs .15, household income .14, food insecurity .21, economic hardship .15 , expense worry .22, perceived stress .20, parenting stress .22, maternal depression .23, co-parenting quality .30, romantic relationship quality .22, parent-child activities .17, and spanking .21. The measures that were not available each year had somewhat higher MDES—maternal anxiety (Beck) .24, maternal anxiety (GAD) .28, physiological stress .30, parent-child interactions (.33).

- In the Results section the authors talk about un-adjusted and adjusted significance levels, and "marginal significance". Since the authors planned to do adjustment for multiple testing, they should only report the adjusted test results. I found the current reporting confusing, in some cases where it is not stated in the text whether the finding is for an adjusted or un-adjusted test I found myself wondering whether this result was significant after adjustment. Also, in null hypothesis significance testing, there is no such thing as "marginally significant", so I suggest dropping this phrase. If a finding does not cross the significance threshold after adjustment for multiple testing, it is simply not significant.

We have revised the text to reflect this suggestion. We only discuss findings as significant if $p < .05$ after adjustments for multiple testing.

- I am not sure what MDES refers to. It should be explained in text. I suspect it is minimally detectable effect size.

We have edited this to spell out “minimally detectable effect size”

- The authors say “To explore whether estimated ITT impacts are equivalent to 0, we conducted two one-sided tests using our regression framework (Lakens, Scheel & Isager, 2018), in which the MDES was set to .20 and .14 (approximately the smallest effect size detectable in our single wave and pooled models respectively).” These equivalence tests are usually conducted against a smallest effect size of interest (SESOI) threshold, not against the minimally detectable effect size. In an underpowered study, it is easily possible that an intervention would turn out to have an effect equivalent to zero according to the minimally detectable effect, but this would not be the case against the smallest effect of relevance/interest. And it could be true the other way around in a highly powered study. So I suggest setting up SESOI thresholds for each outcome based on what effect would still be meaningful for the population of interest.

We respond to this comment below to explain why we have taken out the equivalence test, and also why we do not use SESOI thresholds as suggested by this comment.

- Please give further detail about how the equivalence tests were conducted, so that it is easy to reproduce. I suggest sharing analysis code and data, so that analytical reproducibility can be tested, and the research community can benefit from the hard work of putting together this complex analysis pipeline.

We had calculated the expected raw score that corresponded to the .2 and .14 effect sizes for each measure, and use the “tost” command in STATA to conduct the test. However, we have removed this analysis and table based on your comments below.

- I would not use formal equivalent testing in this study, because it was not pre-registered. I would simply interpret the effect estimates and dispersion statistics as exploratory findings in comparison to SESOI bounds set by the authors. That does not require formal statistical testing, and the results should be interpreted cautiously as exploratory findings that should be confirmed in a confirmatory study.

We appreciate your suggestions about how to think about and (whether to) present the equivalence testing. We had not initially planned to include this analysis in the paper, but we were instructed that the journal requires this to be done for any paper that reports/interprets null results. Of course, that is understandable given how some fields of research have over-relied on p-values for interpreting findings. However, we agree with your comments that what is most important is whether the point estimate falls within a confidence interval that would suggest it could be larger than the smallest effect size of interest SESOI (if there was greater precision/statistical power). Unfortunately, as we first considered implementing the equivalence tests, like many others, we confronted the problem that there really is not an agreed upon effect size change/difference for these outcomes that is considered substantively meaningful. It is our educated *opinion* that the effect sizes differences we see (which are not only small but also in the unexpected direction given our hypotheses) are small enough to not be substantively important, but we also note that this is our opinion rather than consensus in the field. For this reason, our approach is to be as transparent and clear about the magnitude of the findings and our statistical power (and our interpretation of the findings), so that readers can reach their own conclusions. We have taken out the equivalence testing altogether based on these comments and discuss the possibility of substantively meaningful findings too small to be detected in this study in the discussion section.

- The modeling approach seems a bit outdated. I understand that the authors would like to stick to the preregistered analysis approach, but it might be good to conduct a linear mixed effect model as a robustness check with at least family ID and state of residence as random effect predictors. (I would like to again emphasize the importance of sharing analysis code and data within the paper.)

We have estimated the MLM model you suggest with our primary outcomes and the results are below. The results are comparable (in fact, exceedingly close) to those presented in Main paper Table 3 for the OLS pooled sample without any multiple adjustment. As a result, we decided not to change the model we were presenting. (Although if you or the editor or reviewer feels strongly about this, it we could do it). (Implementing the WY adjustment for the MLM model with a sufficiently large number of bootstraps is computationally intensive and thus takes A LOT more time to run).

Summary of ITT Estimates of the Impacts of the BFY High-Cash Gift on Family Wellbeing and Family Processes Measures with Linear Mixed Model for Primary Outcomes (See Main Paper Table 3).

	Effect (Con. Interval)	Std. Effect	ICC Lv. 2 (Lv. 3)	N (Pooled Obs.)	p-value
Panel 1: Economic Resources					
Income-to-needs ratio (including the BFY gift)	3.07 (.52 – 5.62)	.12	.44 (.01)	957 (1844)	.02
Household Income with gift (in 2019 dollars)	.11 (.03 – .20)	.13	.40 (.00)	957 (1844)	.01
Panel 2: Economic Pressure					
Food Insecurity Index	.13 (-.06 – .31)	.08	.48 (.00)	973 (2770)	.17
Non-food Economic Hardship Index	.05 (-.07 – .16)	.05	.43 (.00)	973 (2773)	.42
Expense Worry	.13 (-.03 – .30)	.08	.44 (.01)	973 (2768)	.12
Panel 3: Maternal Psychological Distress					
Perceived Stress Index	.65 (-.00 – 1.30)	.09	.46 (.03)	974 (2771)	.05
Parenting Stress Index	.50 (.11 – .89)	.15	.54 (.01)	965 (1847)	.01
PHQ8 Index of Maternal depression	.20 (-.21 – .61)	.05	.45 (.03)	974 (2768)	.34
Maternal Anxiety (GAD-7)	.26 (-.17 – .69)	.06	.44 (.01)	957 (1840)	.24
Maternal Anxiety (Beck)	.84 (.02 – 1.65)	.13	.49 (.03)	968 (1849)	.04
Physiological Stress (Ln Hair Cortisol)	.04 (-.26 – .33)	.03	.06	364	.81
Panel 4: Interparental Relationship Quality					
Co-Parenting Quality Index	-.32 (-.72 – .08)	-.11	.73 (.02)	803 (1383)	.12
Relationship Quality Index	-.51 (-.91 – -.11)	-.15	.40 (.01)	901 (1877)	.01
Ever cut/bruised/seriously hurt by partner	-.00 (-.03 – .02)	.02	.15 (.00)	771 (1083)	.79
Frequency of Arguing	.05 (-.08 – .17)	.06	.39 (.00)	767 (1078)	.48
Panel 5: Parenting Quality					

Parent-Child Activities Index	.45 (.15 – .74)	.15	.40 (.01)	972 (2763)	.00
Parent-Child Interaction (PICCOLO)	.52 (-.40 – 1.44)	.09	.00	543	.27
Spanking discipline strategy Indicator	-.02 (-.06 – .01)	-.06	.34 (.01)	960 (2427)	.17

Note: Each column presents the raw treatment effect with confidence intervals in parentheses; the standardized treatment effect size; the intraclass correlation (ICC) at the level 2 with ICC at the level 3; number of individuals and observations; and the p -values. The ITT estimates come from the linear mixed model with each age (lv. 1) nested within each child (lv. 2) nested within each state (lv. 3), fitted by the restricted maximum likelihood approach. The model controls for baseline covariates, child age at interview, and phone interview status as fixed effects. Outcomes were standardized using the standard deviation of the low-cash gift within each age. The p -value comes from analyses that do not correct for multiple outcomes. Household incomes across all years are inflation-adjusted to 2019 dollars, and the poverty line is based on the 2019 U.S. Census poverty threshold. Income-to-needs is the household income divided by the poverty line for a given family size and composition. Income and income-to-needs have been truncated at the 99th percentile.

Our responses appear in italics after the reviewer comments.

“Reviewer #2 (Remarks to the Author): Thank you for the comprehensive response and revision, I have no further comments.

We are pleased the reviewer was happy with our revisions.

Reviewer #3 (Remarks to the Author):

The authors have done a good job of responding to my comments and I'm now happy to support publication.

We are pleased the reviewer was happy with our revisions.

Reviewer #4 (Remarks to the Author):

I am grateful to the authors for their diligent work in addressing my comments and the comments of the other referees. I understand that this must have taken quite some time and effort, but it payed off. The manuscript is now significantly improved, and all of my previous comments have been addressed to my satisfaction.

I am glad we have addressed this reviewer's prior comments.

The analysis code attached to the submission is very thorough and well documented. I especially like that it has a very detailed readme file that documents the contents of all code segments and provides instructions on how to replicate their results. (I need to say that I don't use stata, so I am unable to do a computational reproducibility check, but based on the thorough documentation, I have confidence that that would check out as well.)

I am pleased this reviewer has confidence in our coding files.

One remaining minor concern is that the link to the analysis code included in the Code Availability section leads to a repository with multiple folders apparently containing supplementary data for different publications, and I was not able to figure out which one contains the relevant code for this particular paper. It would be important for the link to lead to the relevant file or folder immediately, to make sure that the reader can access the analysis code for this particular study.

We have revised this and now provide the direct link to the code as reference 75.

Since the authors revised some aspects of the manuscript related to statistical power, the editor asked me to pay special attention to issues related to power, so below are my detailed assessment of the sections related to this issue:

Overall I would say that the author's approach and revised discussion section is reasonable. I find it quite transparent, and the conclusions they draw seem to be accurate and not exaggerated. Based on reading the paper, looking at the reported results and the

thoroughness of the report and analysis I largely agree with the conclusions and the wording of the discussion.

I am glad that the reviewer agrees with our wording and conclusions.

In this revised version the authors are quite transparent in that their preregistration is public, and also in their reporting of the sample size rationale. In the previous version of the manuscript they provided a post-hoc justification for the sample size and the power, which built on results that came out after their initial power analysis and sample size target was set. In this post-hoc analysis they refer to McGuire et al 2022 who found that studies which provide an increase of 20% income usually result in an effect size of $d = 0.13$ on average. Adding that since their own study also used a 20% income increase, their minimally detectable effect is in the right ballpark. I criticized this statement, since the income increase was later empirically measured by the authors at 11%. To address my criticism, the authors took this statement out and revised it to be more accurate, and they no longer use this statement to imply that their study was adequately powered. When assessing the manuscript as a whole I find it transparent, that the intervention the authors have applied might be expected to have a smaller effect than what the study was capable of easily detecting. I find the statements of the authors that are currently in the revised manuscript and its discussion section to be accurate and not misleading.”

I am glad the reviewer finds our paper transparent and accurate.

There is one notable exception: "...and our pooled estimates have sufficient statistical power to detect small effects sizes of approximately .14.", first of all, it would be better to use the word "sensitivity" instead of statistical power in this sentence, because power is determined pre-measurement, and once the data has been collected, we usually refer to post-hoc calculated "power" as sensitivity of the design instead. This slight modification in wording could be applied to the relevant sentences throughout the manuscript. This is a minor thing that statisticians would pick up but that will probably not make a difference for 99% of the readers.

We have revised this sentence to use the term "sensitivity" as suggested by the reviewer and have also changed the language throughout the paper where applicable.

A bigger issue with this sentence is the number .14, which is the lowest number they found in their analysis, but for some of the outcomes this number was as high as .33. I would find it more reasonable to use the median in this section as an approximate sensitivity, which was .22 as reported in the Statistical power section.” And relatededly “One thing I noticed during the re-reading is that the numbers related to sensitivity mentioned in the discussion don't quite match up with the numbers reported in the statistical power section: Statistical power section: "This post-hoc approach to statistical power indicates that across the measures, our minimum detectable effect sizes range from .14 to .33, with the median of .22 across all 18 outcomes." Discussion section: "Based on Bloom (1995) for our pooled analysis, the minimum detectable effect size ranges from

.14 to .30, with the average of minimum detectable effect size of .23." This should be sorted out for the numbers (and the metrics mentioned: median vs. mean) to be consistent.

We have made this revision as requested and corrected to refer to the median effect size which is .22. The average MDE was .23 so our text was correct, but we agree with the reviewer that we should be consistent so have revised the text to only mention the median MDE of .22.

As I mentioned, the manuscript in its entirety is sufficiently transparent for my taste. When looking at the discussion section on its own, I still find it reasonable with the revised limitations included in the latest version of the manuscript. Maybe the abstract could be revised to reflect the deficiency in the range of detectible effects. For example in the abstract it could be added to the end of the conclusions that "... but the possibility of very small effects are not ruled out by our results.

We have made this revision to our abstract as requested.